# Tropical cyclones act to intensify El Niño

Qiuyun Wang [1], Jianping Li [2,3], Fei-Fei Jin[3], Johnny C.L. Chan [4], Chunzai Wang[5], Ruiqiang Ding[6], Cheng Sun [1], Fei Zheng[6], Juan Feng[1], Fei Xie[1], Yanjie Li[6], Fei Li[7] & Yidan Xu[1]

Tropical cyclones (TCs), some of the most influential weather events across the globe, are modulated by the El Niño–Southern Oscillation (ENSO). However, little is known about the feedback of TCs on ENSO. Here, observational and modelling evidence shows that TC activity in the southeastern western North Pacific can affect the Niño-3.4 index 3 months later. Increased TC activity in July–September can significantly contribute to the intensity of ENSO in October–December by weakening the Walker circulation and enhancing eastward-propagating oceanic Kelvin waves in the tropical Pacific. Thus, the greater the accumulated cyclone energy, the stronger (weaker) the El Niño (La Niña). A new physics-based empirical model for ENSO is constructed that significantly outperforms current models in predicting ENSO intensity from July to December and addressing the problem about the target period slippage of ENSO. Results suggest that TCs may provide significant cross-scale feedback to ENSO.

[1] College of Global Change and Earth System Sciences (GCESS), Beijing Normal University, 100875 Beijing, China. [2] Key Laboratory of Physical Oceanography/Institute for Advanced Ocean Studies, Ocean University of China and Qingdao National Laboratory for Marine Science and Technology, 266100 Qingdao, China. [3] Department of Atmospheric Sciences, University of Hawai'i at Mānoa, Honolulu, HI 96822, USA. [4] School of Energy and Environment, City University of Hong Kong, Hong Kong, China. [5] State Key Laboratory of Tropical Oceanography (LTO), South China Sea Institute of Oceanology, Chinese Academy of Sciences, 510301 Guangzhou, China. [6] State Key Laboratory of Numerical Modeling for Atmospheric Sciences and Geophysical Fluid Dynamics (LASG), Institute of Atmospheric Physics, Chinese Academy of Sciences, 100029 Beijing, China. [7] Department of Lower Atmosphere Observation Research (LAOR), Institute of Atmospheric Physics, Chinese Academy of Sciences, 100029 Beijing, China. Correspondence and requests for materials should be addressed to J.L. (email: ljp@ouc.edu.cn)

El Niño–Southern Oscillation (ENSO) is the leading source of interannual climate variability, influencing weather and climate over the globe[1–4]. However, owing to initial errors in models and random atmospheric disturbances[5–9], ENSO predictability remains a key challenge[10], particularly its intensity[11–13] and the target period slippage of ENSO[14]. Tropical cyclones (TCs), as the most influential weather events globally[15–17], have also attracted wide attention[18–23]. Numerous studies have shown that ENSO modulates TC activity[24–30]. TCs in the tropical Pacific might have led to an El Niño-like warming in the tropical Pacific during the early Pliocene epoch[31]. In modern times, >30% of TCs worldwide occur in the western North Pacific (WNP), resulting in enormous societal impacts on littoral regions. More TCs tend to be generated in the southeastern WNP[26] during El Niño developing years (see Methods). However, the effect of TCs on El Niño intensity has not been examined on interannual timescales. The motivation of this study is to explore the role of the southeastern (10°−20°N, 135°−170°E) WNP TC (in short, WNP TC) activity in El Niño events from the perspective of accumulated cyclone energy (ACE)[32]. We also consider whether TC activity can be used for ENSO forecasting. Good prediction skill will provide powerful evidence for the role of WNP TCs in ENSO.

Current study demonstrates that the ACE index in the region (10°−20°N, 135°−170°E; hereafter WNP ACE) leads the Niño-3.4 index (N3.4) by about 3 months, as deduced from observations and an intermediate-complexity coupled ocean–atmosphere model[11,33]. WNP TCs in July–September (J–A–S) can significantly intensify El Niño in October–December (O–N–D); and the greater the ACE, the stronger the El Niño. WNP TCs weaken significantly the Walker circulation by stressing lower-level anomalous westerlies over 0°−10°N in the tropical western–central Pacific and generating a Hadley-like circulation in the tropical western Pacific. TCs also shallow the tropical western Pacific thermocline and enhance eastward-propagating Kelvin waves, resulting in a significant decrease of zonal thermocline gradient in the equatorial Pacific. Consequently, El Niño is intensified. In same way, La Niña is weakened; and the greater the ACE, the weaker the La Niña. In addition, this study constructs a new physics-based empirical model (the ACE + N3.4 model) for ENSO. This new model is significantly better than the current dynamical and statistical models for predicting ENSO intensity from July to December and addressing the target period slippage of ENSO as it predicts ENSO maxima closer to the time they are actually observed. The ACE + N3.4 model can successfully predict the intensity of extreme El Niño in 2015, and it does not show any El Niño signal before October in 2014, instead of a strong or even extreme El Niño to occur later in 2014 as many models' prediction.

## Results

**The El Niño becomes more intense when the ACE is greater.** There are larger ACE anomalies over the southeastern WNP in El Niño developing years (Supplementary Fig. 1), associated with a southeastward shift of the mean TC genesis location[27,34]. We find that the WNP ACE leads the N3.4 by ~3 months (see Supplementary Fig. 2). The correlation of the N3.4 with its value 3 months earlier is closely related to the WNP ACE of 3-months earlier during El Niño developing years (see Supplementary Fig. 3 and Supplementary Table 1). The greater the ACE, the stronger the El Niño, particularly for TCs in J–A–S. As seen from Fig. 1a (combined with Supplementary Fig. 4,a–e), the mean explained percentages (see Methods) of both the WNP ACE and N3.4 in J–A–S to N3.4 in O–N–D are ~51%. The mean explained percentage of the N3.4-independent ACE in J–A–S to the N3.4 in O–N–D is equivalent to that of the ACE-independent N3.4

(17.3% vs 18%, respectively). The joint explained percentage of the preceding (3 months earlier) ACE and N3.4 is nearly 20% higher than that of either factor alone. An intermediate-complexity coupled ocean–atmosphere model[11,33] can predict the sea-surface temperature (SST) anomalies pattern well during El Niño developing years. Both the time series of N3.4 and the spatial patterns of the SST anomalies from observations and simulations are similar. The correlation coefficient between observed and simulated N3.4 reaches 0.86 (Supplementary Fig. 5). Model results agree with the aforementioned observations (Fig. 1b and Supplementary Fig. 4,f–j). The J–A–S WNP ACE can significantly intensify the O–N–D Niño-3.4 SST anomalies during El Niño developing years (Fig. 2a), whereas the Niño-3.4 SST anomalies associated with simultaneous ACE (Figs. 2b, c) are noticeably weaker. These results support the finding that the J–A–S ACE has an important influence on the O–N–D El Niño. In addition, unlike N3.4, the autocorrelation of the WNP ACE with its value 3 months earlier is not significant (Supplementary Fig. 2a), and the ACE still leads the N3.4 even after removing the simultaneous N3.4 and the autocorrelation of the N3.4 index with its value 3 months earlier (Supplementary Fig. 2b). These results indicate that the lead of WNP ACE index over the N3.4 does not primarily result from the autocorrelation of the N3.4. Therefore, the cross-scale effect of WNP TC activity on El Niño events is not negligible. The mechanism underlying the effects of WNP TCs on El Niño is discussed next.

**Influence of daily WNP ACE on westerly anomalies.** Taking the year 2015 as an example, as shown in Fig. 3a, daily strong westerly (easterly) anomalies are found on the southern (northern) flanks of TCs. From 1970 to 2016, the mean value of TC-related anomalous westerlies (5.41 m s$^{-1}$) is 1.81 times that without TC occurrence (2.99 m s$^{-1}$), and this ratio holds regardless of whether it is an El Niño developing year. This suggests that the enhancement of westerly winds related to TCs is independent of El Niño occurrence. The westerly anomalies associated with TC occurrence shift northward as TCs move northward. The range of latitudes of anomalous westerlies affected by TCs reach 5°−10° south of the TC centre. The regional meridionally averaged zonal wind anomalies on the southern flanks of TCs in 2015 are shown in Fig. 3b. There are stronger anomalous westerlies when TCs are present. The mean value of TC-related anomalous westerlies (4.94 m s$^{-1}$) is 2.66 times that without TC occurrence (1.86 m s$^{-1}$) from 1970 to 2016, and this ratio changes little in El Niño developing years. In addition, the centres of strong TC-related westerly anomalies shift westward as TCs move westward. Because the low-pressure zones resulting from TC genesis do not vanish immediately as the TC moves away[35], a TC's influence on anomalous westerlies remains after the TC moves away or disappears (Fig. 3c). These features indicate that TC occurrence intensifies the westerly anomalies on the southern flanks of TCs. Anomalous wind forcing related to TCs excites or amplifies Kelvin waves, further affecting the Niño-3.4 SST anomalies[36].

**Interannual climatic effect of the WNP ACE on El Niño.** On interannual timescale, at 850 hPa (Fig. 3d), WNP TCs can lead to anomalous westerlies over 0°–10°N in the tropical western–central Pacific. This might result from tropical semi-geostrophic adjustment processes (see Supplementary Fig. 6a), as more intense WNP TCs lead to a broad area of low pressure in the WNP, and the centre of significant zonal wind anomalies is located south of the selected ACE region (this centre is consistent with the main effect of the mean WNP TC genesis position at 12.07°N, 155.36°E). The anomalous westerlies increase the Niño-3.4 SST anomalies. At 200 hPa (Fig. 3e), anomalous easterlies

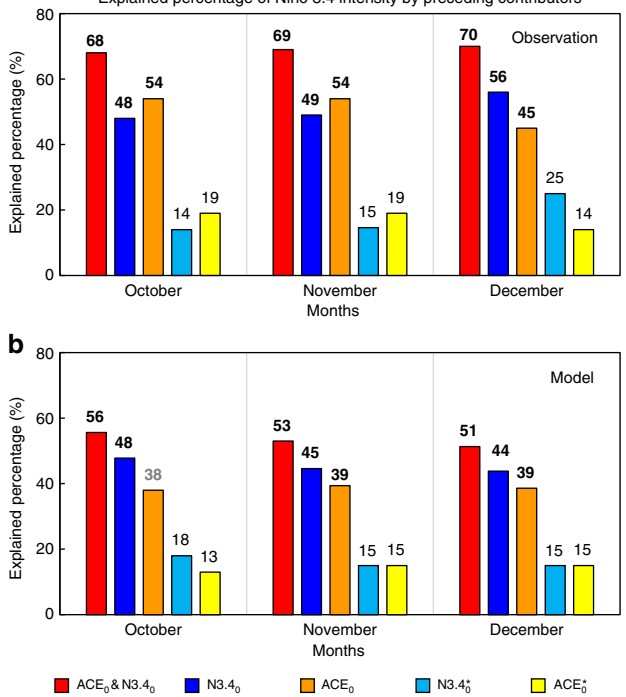

**Fig. 1** Explained percentage (%) of the running 3-month mean SST anomaly for the Niño 3.4 region (5°N−5°S, 120°−170°W) (N3.4, an ENSO index) from October–December obtained using the preceding (3-months earlier) contributors' signals during El Niño developing years (1970−2016). **a** Observations. **b** Simulated results from the intermediate-complexity coupled ocean–atmosphere model[11,33]. An El Niño developing year is defined when the above-moderate El Niño events (including moderate events) develops from weak to strong. $ACE_0\&N3.4_0$ (red bars) represents the combined contribution of preceding accumulated cyclone energy (ACE) over the western North Pacific (10°–20°N, 135°–170°E; WNP) and N3.4; $N3.4_0$ (blue bars) the preceding N3.4; $ACE_0$ (orange bars) the preceding WNP ACE; $N3.4_0^*$ (sky blue bars) the preceding ACE-independent N3.4 (i.e. the preceding N3.4 after removing WNP ACE); and $ACE_0^*$ (yellow bars) the preceding N3.4-independent ACE index (i.e. the preceding WNP ACE after removing N3.4). Black (grey) bold font indicates statistical significance above the 95% (90%) confidence level (Student's $t$-test)

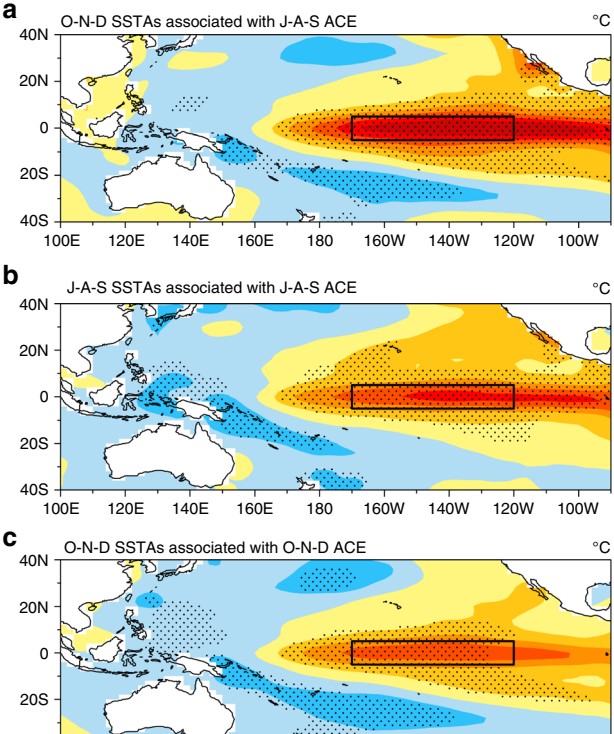

**Fig. 2** Tropical Pacific sea-surface temperature (SST) anomalies (°C) associated with the accumulated cyclone energy (ACE) anomalies over the western North Pacific (10°–20°N, 135°–170°E; WNP) in the period 1970–2016. **a** Composite of the O–N–D (each month from October to December) SST anomalies (°C, shading) associated with the J–A–S (each month from July to September) WNP ACE anomalies during the El Niño developing years. An El Niño developing year is defined when the above-moderate El Niño events (including moderate events) develops from weak to strong. **b** Composite of the J–A–S SST anomalies (°C, shading) and the J–A–S WNP ACE anomalies. **c** Composite of the O–N–D WNP SST anomalies (°C, shading) and the O–N–D WNP ACE anomalies. The stippled regions denote statistical significance above the 95% confidence level (Student's $t$-test). The black rectangle denotes the Niño-3.4 region (5°S–5°N, 120°–170°W)

resulting from TCs occur from 10°N to 10°S, which do not seem to conform to the ACE locations. The main reason for this is a Hadley-like circulation (20°N−20°S, 135°−170°E) enhanced by the TCs (Fig. 3f) on the basis of tropical semi-geostrophic adjustment processes at 200 hPa (see Supplementary Fig. 6b). Anomalous northerlies south of the equator enhance the symmetry of the wind field in the upper troposphere with respect to the equator (see Supplementary Fig. 6c), intensifying the anomalous upper easterlies over the tropical Pacific. Both the direct effect of asymmetrical anomalous westerlies at lower levels and the indirect effect of the Hadley-like circulation weaken the Walker circulation (5°N−5°S, 120°E−120°W; the Walker index changes from −5.47 to −3.84 m s$^{-1}$; see Methods) over the equatorial region in J–A–S, further enhancing the equatorial Pacific eastward-flowing currents (see Supplementary Fig. 6d), and thereby increasing the Niño-3.4 SST anomalies. The eastward-flowing currents are located north of the equator in J–A–S, which is consistent with lower-level anomalous westerlies. The ACE leads the lower-level westerlies (Walker circulation) in the tropical western–central Pacific by 2 (3) months regardless of whether the proceeding westerlies (Walker circulation) or their simultaneous signals are removed (see Supplementary Fig. 7).

This lead–lag correlation between the ACE and circulation is in accordance with that between the ACE and N3.4 (see Supplementary Fig. 2). Through the effect on the Walker circulation, the Niño-3.4 SST anomalies associated with TCs are strongest after 3 months.

The evolution of the horizontal wind anomalies is also examined (see Supplementary Fig. 8a−c). Compared with the situation in J–A–S, the significant anomalous westerlies related to TCs at 850 hPa shift eastward and become symmetric about the equator in O–N–D, and are accompanied by a significantly weakened Walker circulation. At the upper tropospheric level, the centre of significant anomalous easterlies also shifts eastward. All these effects enhance the eastward-flowing currents, and the centre of the eastward-flowing currents shifts equatorward in O–N–D (see Supplementary Fig. 8d), and the Niño-3.4 SST anomalies significantly increase.

Another possible mechanism is the change in thermocline depth associated with WNP TCs. In J–A–S, TCs can shallow the tropical western Pacific thermocline (Fig. 4a). Agreeing with theory, the observed thermocline is shallowed via Ekman pumping associated with the downdrafts over 0°−10°S. However,

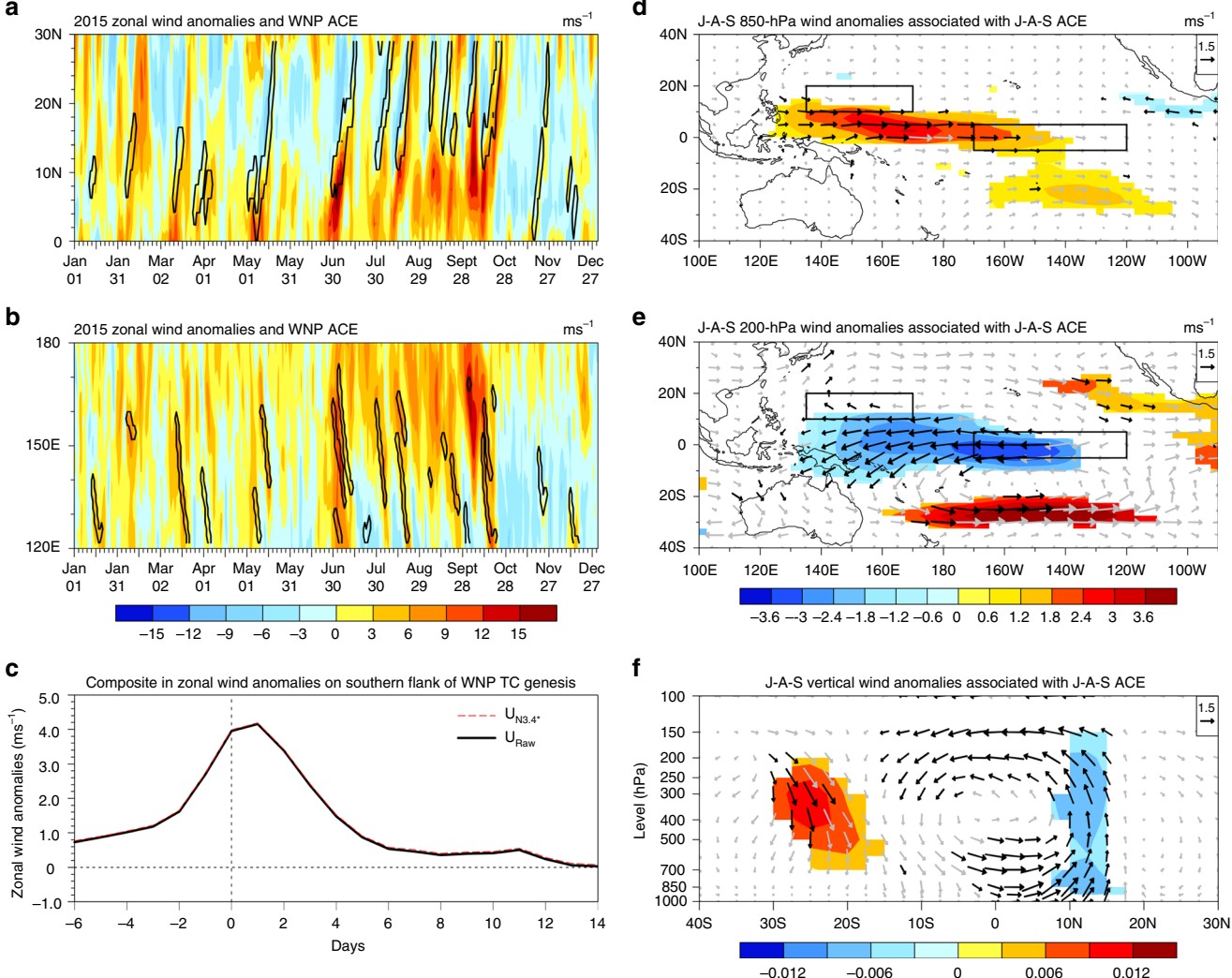

**Fig. 3** Wind anomalies and circulation associated with the accumulated cyclone energy (ACE) anomalies over the western North Pacific (10°–20°N, 135°–170°E; WNP). **a** Latitude–time Hovmöller diagram (along 135°–170°E) of zonal wind anomalies (shading, m s$^{-1}$) in 2015. Black contours indicate the 1 m$^2$ s$^{-2}$ isoline of WNP ACE, representing the TC life cycle. **b** As in **a**, but for longitude–time (along 0°–15°N). **c** Composite time series of zonal wind anomalies (m s$^{-1}$) on the southern flank (within 5° of lon$_0$, and within 10° south of lat$_0$; lon$_0$ and lat$_0$ indicate the longitude and latitude of the TC genesis position, respectively) of single TC genesis in the period 1970–2016. Black (red) line indicates the wind anomalies with (without) the contribution of the Niño-3.4 SST. **d** Composite of the 850-hPa zonal (shading) and horizontal (vectors) wind anomalies (m s$^{-1}$) in J-A-S associated with the J-A-S WNP ACE during El Niño developing years. An El Niño developing year is defined when the above-moderate El Niño events (including moderate events) develops from weak to strong. Shading and black vectors indicate significance above the 95% confidence level using Student's *t*-test. The two black rectangles denote the selected ACE region (10°–20°N, 135°–170°E) and the Niño-3.4 region (5°S–5°N, 120°–170°W). **e** As in **d**, but for the 200-hPa level. **f** As in **d**, but for the vertical *p*-velocity (shading, Pa s$^{-1}$) and the wind (vectors) in the vertical–meridional plane. Vectors are obtained by the zonal wind anomalies (m s$^{-1}$) and magnified vertical *p*-velocity (×(−200))

there are updrafts over 5°–10°N, so theoretically the thermocline in these latitudes should deepen, contrary to what is actually found. One possible reason for this discrepancy is that wind forcing related to TCs excites or amplifies Kelvin waves, further affecting tropical thermocline adjustments[36,37]. Another is TC turbulent mixing processes, in which strong surface TC winds cool the surface and warm the subsurface waters[38–42]. This vertical mixing might have a thermodynamic effect on wave formation. Warm water in the tropical western Pacific is carried eastward, in association with enhanced eastward-propagating equatorial Kelvin waves, further deepening the tropical eastern Pacific thermocline and thereby reducing the equatorial Pacific zonal thermocline gradient (Fig. 4c). Due to the weakened Walker circulation, the thermocline at 5°S–15°N in the tropical western Pacific is shallower and flatter in O–N–D than in J–A–S, and the

equatorial Pacific zonal thermocline gradient becomes smaller (Fig. 4b, d), as the Niño-3.4 SST anomalies become stronger. To further investigate the ocean-process effects of WNP ACE on El Niño, the depth–zonal distribution of monthly equatorial potential temperature anomalies averaged between 5°S–5°N during the El Niño developing year is shown in Supplementary Fig. 9. At the beginning of the year, the positive potential temperature anomalies begin to emerge in the western Pacific and then propagate eastward along the main thermocline and accumulate in the equatorial eastern Pacific. From June, the negative potential temperature anomalies grow in the western Pacific because of the eastward transport of warm surface water and upward compensation by cold water, which further enhances positive potential temperature anomalies in the equatorial eastern Pacific by transporting warm surface water. After removing the

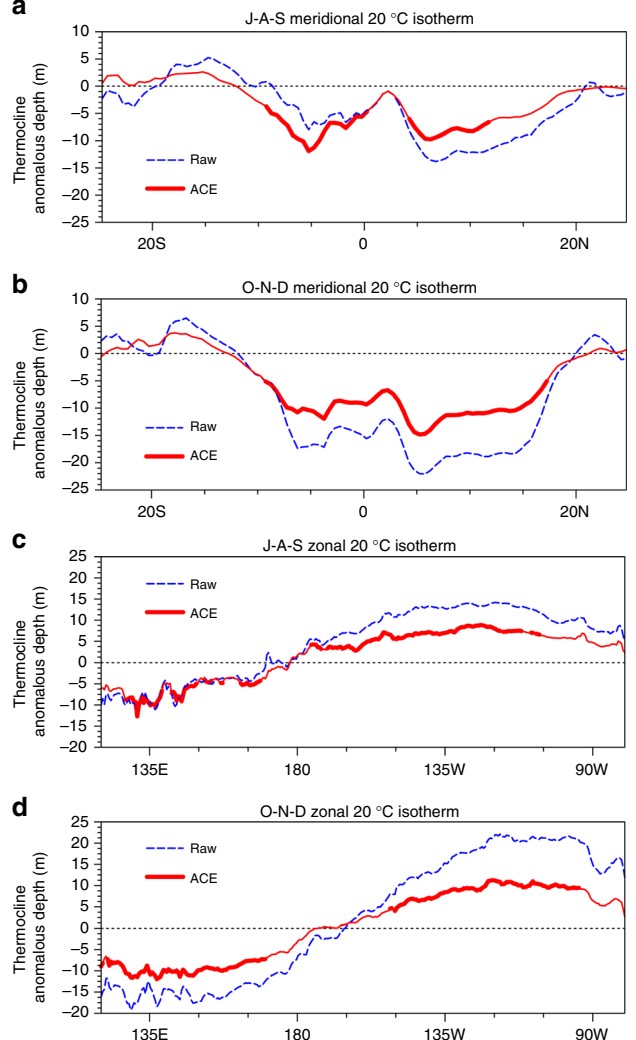

**Fig. 4** Thermocline (20 °C isotherm depth, m) during El Niño developing years (1970–2010). **a** Meridional distribution (along 135°–170°E) in J-A-S. **b** As in **a**, but for O-N-D. **c**, Zonal distribution (along 5°S–5°N) in J-A-S. **d** As in **c**, but for O-N-D. The blue dashed line indicates the original thermocline; the red line associated with the preceding WNP ACE signal in J-A-S, and the thicker line indicates significance above the 95% confidence level. An El Niño developing year is defined when the above-moderate El Niño events (including moderate events) develops from weak to strong

simultaneous TC signals (Fig. 5a), this change in the potential temperature anomalies decreases, and can even disappear, particularly during the WNP TC season (May–October; Fig. 5b). In addition, when the proceeding ACE signal is removed, the potential temperature anomalies also decrease, and can even disappear, particularly in the 3 months after the TC season (Supplementary Fig. 10). All the studied observations indicate that WNP TCs can enhance the transport of warm water from the equatorial western-to-eastern Pacific, thus increasing the Niño-3.4 SST anomalies.

**Role of the Madden–Julian Oscillation (MJO).** Previous studies[43,44] indicated the MJO plays an important role in the onset of El Niño and can also affect WNP TC activity on intra-seasonal timescale. Hence, the role of the MJO in the modulation of TCs to the El Niño intensity is investigated. The lead–lag correlation between WNP ACE and SST is largely unaffected by removal of the MJO signal, and the magnitude of the correlation

coefficient only changes slightly (Supplementary Fig. 11). When compared with the original time series, the intensity of N3.4 changes only slightly by removing the MJO signal. This is also true for the WNP ACE. In addition, the intensity of the N3.4 related to the preceding ACE (after removing the MJO signal) is also only slightly changed (Supplementary Fig. 12). Thus, the MJO signal might only slightly affect the modulation of TCs to ENSO intensity on the interannual timescale. In addition, all three types of the MJO conditions (active MJO, inactive MJO, and non-MJO events) occur in J-A–S during eight studied El Niño developing years (Supplementary Fig. 13). There are no significant differences among the occurrence frequency of the three types, and the non-MJO events occur most often, particularly in July (Fig. 6). This suggests that the modulation of TCs to El Niño intensity on an interannual timescale has few dependencies on the MJO conditions. To further verify this conclusion, hit rates (see Methods) between ACE and MJO events in El Niño developing years and a composite year are investigated. Monthly hit rates remain low for both individual years and the composite year (21.8% for July, 35.4% for August and 30.8% for September), except for 2015. Results indicate the MJO does not play a major role in modulating TCs to El Niño intensity.

**A new physics-based empirical model for ENSO prediction.** In a La Niña event, the key domain of the preceding WNP TCs changes to 10°−25°N, 130°−155°E and WNP TCs have a negative effect on La Niña (Supplementary Fig. 14). The greater the WNP ACE, the weaker the La Niña. Based on the possible physical mechanisms that relate the preceding WNP TCs to ENSO, a new physics-based empirical model for ENSO (the ACE + N3.4 model) is constructed. The predicted skill of the ACE + N3.4 model for ENSO is assessed by comparison with single-factor models (N3.4 model and ACE model), current dynamics and statistical models.

A holdout method (see Methods) is first employed and the N3.4 predicted by models is compared with observations. There is a high correlation coefficient (~0.91) and sign consistency (92%) between the observed N3.4 and that predicted by the ACE + N3.4 model, much higher than for the single-factor N3.4 model (correlation coefficient of 0.75, sign consistency of 79%) and ACE model (correlation coefficient of 0.66, sign consistency of 65%) (Fig. 7a). The root mean square error (RMSE) between the observed N3.4 and that predicted by the ACE + N3.4 model is 0.39 °C, which is smaller than that for the N3.4 model (0.58 °C) and the ACE model (0.68 °C). These results indicate that the ACE + N3.4 model has better prediction skill for N3.4 than single-factor models. Because the ACE model has the lowest prediction skill, we only consider the ACE + N3.4 model and the N3.4 model in further analyses of El Niño and La Niña events. The N3.4 predicted by the ACE + N3.4 model is closer to observations than that predicted by the N3.4 model during the development periods of El Niño and La Niña events (Fig. 7c, e). The peak N3.4 index obtained with the N3.4 model occurs 3 months after the peak in the observations, while the predictions by the ACE + N3.4 model peak at the same time as the observations. The maximum relative advantage of the ACE + N3.4 model for N3.4 relative to the N3.4 model is about 0.62 °C in November during the El Niño developing year. This value of the maximum relative advantage is about 43% of the observed anomaly in November. Although the predicted advantage of the ACE + N3.4 model relative to the N3.4 model for N3.4 in other months is smaller than in November, the proportion of the relative advantage in observations in some months might be higher than in November, such as in April, May, June and July. The relative predicted advantage of the ACE + N3.4 model is also evident during the La

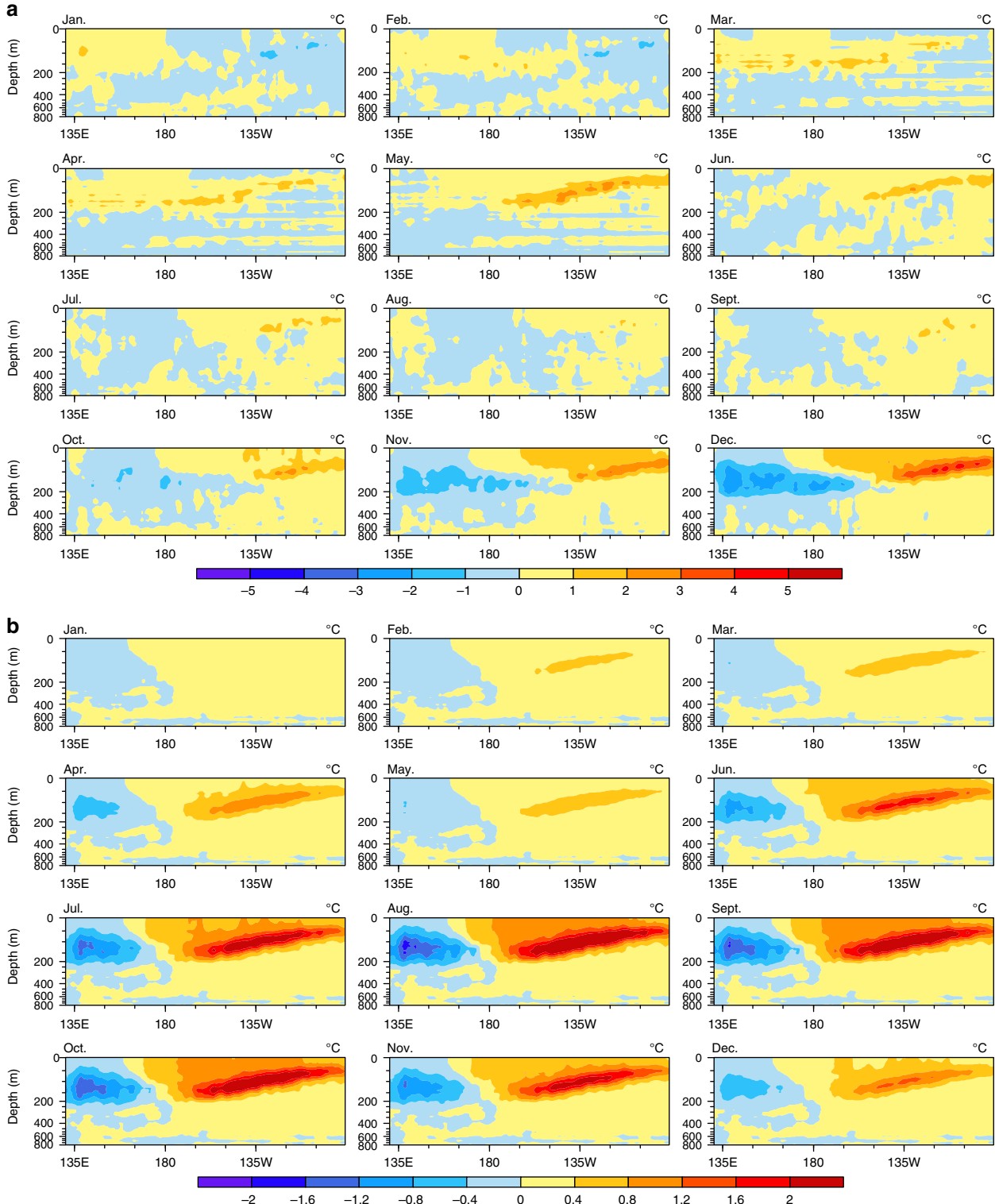

**Fig. 5** Composite of the depth–zonal distribution of monthly equatorial potential temperature anomalies averaged between 5°S−5°N (shading; °C) during the El Niño developing years (2004−2016). **a** Distribution of monthly equatorial potential temperature anomalies after removing the simultaneous WNP ACE. **b** As in **a**, but relating to the simultaneous WNP ACE. An El Niño developing year is defined when the above-moderate El Niño events (including moderate events) develops from weak to strong

Niña developing year, with a maximum of nearly 0.48 °C in December; this value of maximum relative advantage is about 50% of the observed value. This proportion might be higher in October and November. The predictions by the ACE + N3.4 and N3.4 models are similar during the decaying periods of both types

of event, because there are few TCs in this period. To evaluate further the predicted ability, a running holdout method is employed. The prediction skill of the ACE + N3.4 model for N3.4 is better than that with the holdout method (Fig. 7b). It does not seem to be very different. The predicted N3.4 from July to

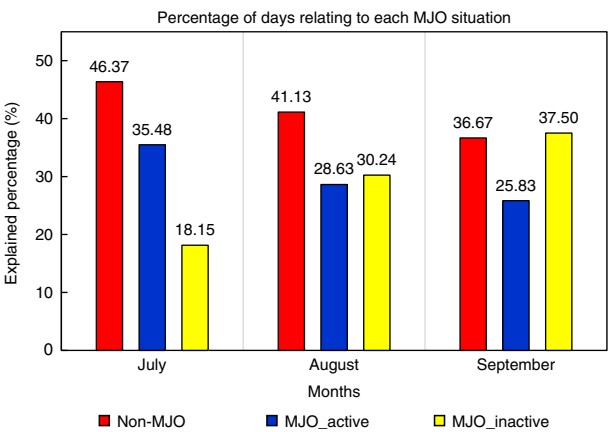

**Fig. 6** Occurrence frequency (%) of three Madden–Julian Oscillation (MJO) event types from July to September during eight El Niño developing years in the period 1979–2016. Red bars represent the occurrence frequency (%) of non-MJO, blue for active MJO, and yellow for inactive MJO. An El Niño developing year is defined when the above-moderate El Niño events (including moderate events) develops from weak to strong

December closely approximates observations because of the TC contribution (Supplementary Fig. 15). The ACE + N3.4 model also shows better predicted ability for ENSO than the N3.4 model, especially during the development periods of El Niño and La Niña events (Figs. 7d, f). This predicted ability does not change with the selection of the training period (Supplementary Fig. 16); the mean correlation coefficient between predicted and observed N3.4 is 0.912 and the mean RMSE is 0.376 °C. This verifies the stability of the ACE + N3.4 model.

Model prediction skills for ENSO are compared for the ACE + N3.4 model, 18 dynamical models and eight current statistical models. For the whole year, the N3.4 predicted by the ACE + N3.4 model shows a high correlation (~0.92; Fig. 8a) and low RMSE (~0.35 °C; Fig. 8b) with observations at a lead time of 2 months (see Methods). The correlation coefficient (~0.91) and RMSE (~0.35 °C) between the average skill level of dynamical models and observed N3.4 are nearly equivalent to those from the ACE + N3.4 model, but the current statistical models clearly show a predicted disadvantage relative to the ACE + N3.4 model; the average correlation coefficient between prediction and observations is ~0.88 and the RMSE is ~0.42 °C. For the period July−December, the N3.4 predicted by the ACE + N3.4 model shows a higher correlation coefficient (~0.94; Fig. 8c) and lower RMSE (~0.32 °C; Fig. 8d) than the average for both the dynamical models (correlation coefficient of ~0.91, RMSE of ~0.40 °C) and pre-existing statistical models (correlation coefficient of ~0.90, RMSE of ~0.47 °C). However, for the period January−June, the ACE + N3.4 model shows a predictability barrier, with its prediction skill for N3.4 being close to the average skill of the pre-existing statistical models (Fig. 8e, f). This might be related to the WNP TC genesis season, as there are few TCs in this period, which further verifies the importance of TCs in ENSO forecasts. El Niño and La Niña events are also examined. Compared with the average prediction skill for current dynamical models (Fig. 9a, b), the ACE + N3.4 model has a higher prediction skill for ENSO intensity during the ENSO developing period. Both the N3.4 predicted by the ACE + N3.4 model and the observed N3.4 show a peak at the same time, while the peaks of average predictions from current dynamical models lag the observations. This predicted advantage of the ACE + N3.4 model for ENSO is more evident when compared with the average prediction skill for current statistical models (Supplementary Fig. 17). The unexpected halt of N3.4 growth in 2014 and the development of an

extreme El Niño in 2015 have attracted much research interest[45]. Here, the predicted N3.4 values in 2014/2015 are shown. Unlike most of the current dynamical and statistical models, the ACE + N3.4 model does not show the El Niño signal before October in 2014 (Fig. 9c) because of the anomalous absence of the preceding TCs (Supplementary Fig. 18a). In 2014, the WNP ACE is far below the average ACE level during El Niño events, and is even close to the average WNP ACE during La Niña events. The N3.4 predicted by the ACE model closely follows the variations in the observations after February of that year (Supplementary Fig. 18b). In 2015, the ACE + N3.4 model successfully predicts El Niño intensity close to the observations because the preceding WNP ACE in 2015 was well above the average ACE level of El Niño events. During the 2015 El Niño developing period, the ACE + N3.4 model shows even some predicted advantage relative to the average skill of the dynamical models.

**Discussion**

In this study, the feedback of WNP TCs on ENSO on interannual timescale is examined using observational data over several decades, revealing that WNP TCs can significantly intensify Niño-3.4 SST 3 months later, particularly for J–A–S TCs. This modulation has few dependencies on the MJO. WNP TCs in J–A–S can affect both the atmospheric and oceanic processes that drive the O–N–D ENSO, which differs from the early Pliocene epoch[18]. As shown in Fig. 10, WNP TCs can weaken the Walker circulation via direct effects of equatorial asymmetrically anomalous westerlies at lower tropospheric levels and indirect effects of Hadley-like circulation. TCs can shallow the thermocline in the tropical Western Pacific. Warm water in the tropical Western Pacific is carried eastward, in association with the enhanced eastward-propagating equatorial Kelvin waves, further deepening the thermocline in the tropical eastern Pacific, thereby reducing the gradient of the zonal thermocline in the equatorial Pacific. These two processes lead to an intensified El Niño or weakened La Niña. The greater the WNP ACE, the stronger (weaker) the El Niño (La Niña). Secondly, a new physics-based empirical model (the ACE + N3.4 model) has been constructed based on the preceding ACE and N3.4. Compared with current dynamical and statistical models, the ACE + N3.4 model, although very simple, gives a significantly better ENSO intensity forecast, especially for June −December during the developing period of ENSO. Moreover, the peak N3.4 predicted by current dynamical and statistical models lags the observations (the target period slippage of ENSO[14]), but for the ACE + N3.4 model the predictions and observations peak at the same time. The TC genesis season in the WNP directly affects the forecasting ability of the ACE + N3.4 model. The predictions of the ACE + N3.4 model also provide powerful evidence that TCs are essential to ENSO development and can significantly improve the prediction skill for ENSO intensity.

Camargo and Sobel[46] also found this SST-lagged signal using ACE lag correlations (July−October) with Niño indices for different seasons, but the maximum correlation between the ACE and N3.4 occurs when N3.4 leads ACE in their study. Thus, they suggested that ACE leads N3.4 primarily because of the auto-correlation of SST anomalies in the Niño-3.4 region, and that TCs only play a small positive role in ENSO dynamics. To examine this result, we further investigate the relationship between the ACE in the key domain (10°−20°N, 135°−170°E) from July to October and the seasonal N3.4 during 1970−2016 and 1970 −2002 (the latter is the period used in Camargo and Sobel's work). Results indicate that the correlation between ACE from July to October and the seasonal N3.4 reaches a maximum when ACE leads N3.4. Our aforementioned analysis indicates that TCs

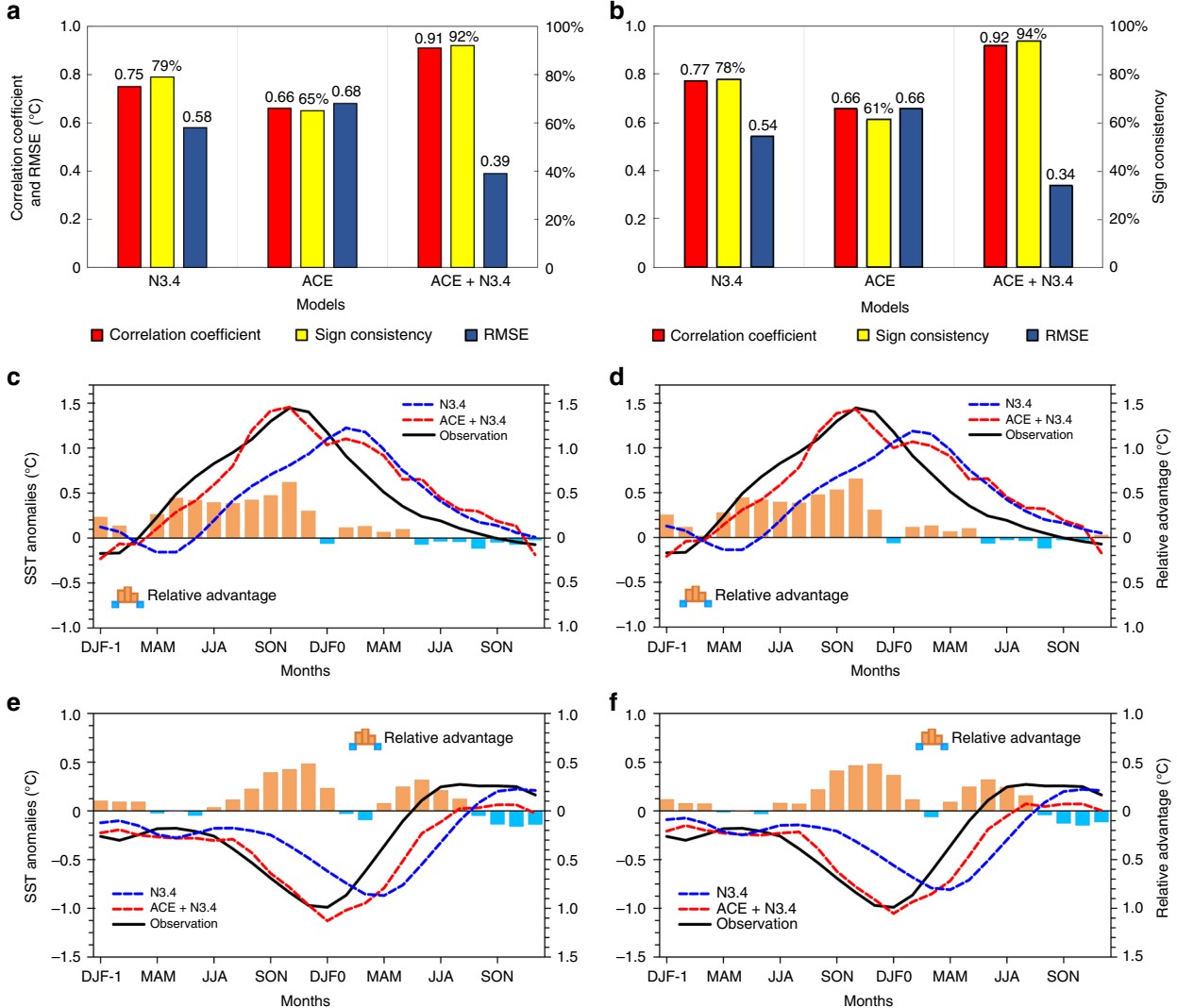

**Fig. 7** Prediction skill of three statistical models for the running 3-month mean SST anomaly for the Niño 3.4 region (5°N−5°S, 120°−170°W) (N3.4, an ENSO index) in the hindcasting period 2000−2016. **a** Correlation coefficient (red bars), sign consistency (%, yellow bars) and root mean square error (RMSE, °C, dark blue bars) between N3.4 (°C) observations and predictions based on three statistical models (N3.4, ACE, and ACE + N3.4) using the holdout method. **b** Same as **a**, but for results using the running holdout method. **c** Composite time series of N3.4 for observations (black solid line) and predictions (dashed lines) during El Niño events (all events, including 2002−2003, 2004−2005, 2006−2007, 2009−2010, 2014−2015 and 2015−2016) with the holdout method. The blue (red) dashed line is for the N3.4 (ACE + N3.4) model. Bars indicate the amplitude of the models' relative advantage (relative advantage$_{\text{model A}\rightarrow\text{model B}}$ = |Model A − Observation| − |Model B − Observation|) between the N3.4 and ACE + N3.4 models. Blue (orange) bars represent the advantage of the N3.4 model relative to the ACE + N3.4 model (the ACE + N3.4 model relative to the N3.4 model). DJF-1 and DJF0 represent the December−February in last year and the year concurring with El Niño, respectively. **d** Same as **c**, but for results using the running holdout method. **e** Same as **c** but for La Niña (all events, including 2000−2001, 2005−2006, 2007−2008, 2008−2009, 2010−2011 and 2011−2012). **f** Same as **e** but for results using the running holdout method

play an important role in the intensity of ENSO. Therefore, our current finding is evidently different from that of Camargo and Sobel.

Although the primary focus here is on the feedback of WNP TC occurrence on ENSO, other related topics warrant further investigation. For example, the effect of the MJO and other factors on TC genesis and the causality associated with the influence of WNP TCs on Hadley-like circulations and tropical westerlies still need to be further verified in a fully coupled model. Previous studies[31,37,42] have provided fundamental insights into simulating climatic TCs in a fully coupled model, but were restricted to TC climatic distributions (or to accumulated TCs). Currently, fully coupled models that perform well in simulating interannual variabilities of both TC activity and El Niño are rare, and remain

an important challenge for researchers. In general, although TC activity is usually considered a synoptic event, the cumulative effect of TCs can provide a significant cross-scale feedback to the large-scale atmospheric and oceanic circulations that then affects the intensity of ENSO. TC activity thus help to improve ENSO forecasts. How this under-investigated relationship between TCs and ENSO is incorporated into existing forecasting models still needs to be determined. In addition, these results highlight the need for further study of the cross-scale effect of short-timescale events on long-timescale events. Perhaps the cumulative effect of TCs in other basins (e.g. hurricanes over the North Atlantic) affects intraseasonal or longer-timescale climatic events in other regions by modulating the large-scale dynamic processes. The cumulative effect may be a critical link between synoptic and

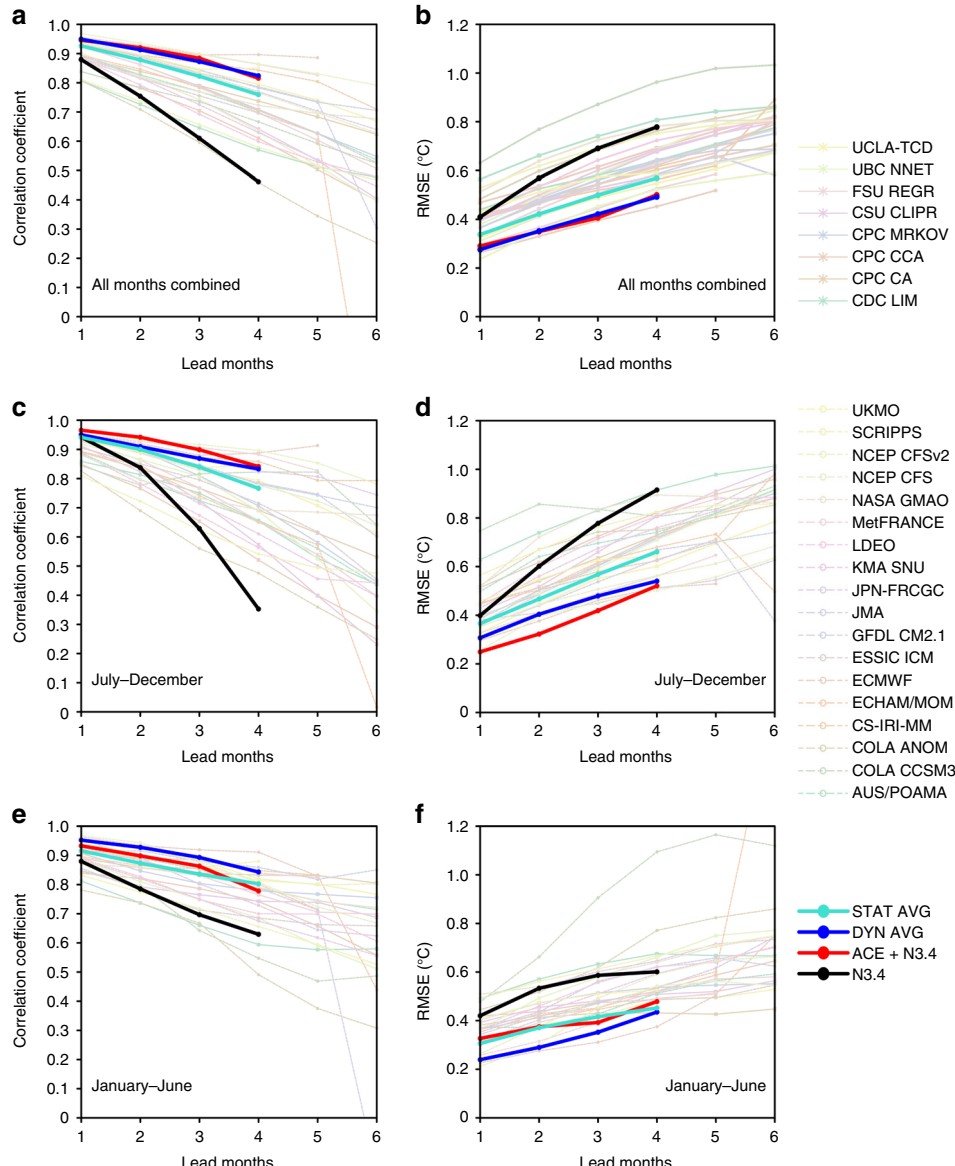

**Fig. 8** Temporal correlation and root mean square error (RMSE, °C) between predictions and observations from 2002 to 2016, as a function of lead time. **a** Correlation coefficient for all months combined. Each line represents one model. Eighteen dynamical models (thin dashed lines with circles), eight statistical models (thin solid lines with asterisks), the N3.4 model (black thick line with dots) and the ACE + N3.4 model (red thick line with dots) are employed. The blue (turquoise) thick line with dots denotes the average of the dynamical (statistical) models. **b** Same as **a**, but for RMSE (°C). **c** Same as **a**, but for July–December. **d** Same as **c**, but for RMSE. **e** Same as **a**, but for January–June. **f** Same as **e**, but for RMSE

climatic events. If this cumulative effect of short-term events can be reproduced well in the existing forecasting models, the prediction skill of models would be improved. Furthermore, we can also apply this cumulative effect to cross-scale studies in other disciplines. For example, in oceanography, it may deepen our understanding of the inverse energy cascade from mesoscale eddies to large-scale circulation. All these aspects belong to the category of cross-scale dynamics, which deserves further in-depth study.

## Methods

**Data pre-processing**. A 3-month running mean is taken of all monthly (but not daily) datasets for the period 1970–2016 (1970–2010 for ocean data). N3.4 is the running 3-month mean Niño-3.4 SST anomalies (5°N–5°S, 120°W–170°W). El Niño and La Niña events are selected by a standard used by NOAA (http://ggweather.com/enso/oni.htm). Generally, El Niño (La Niña) events are defined as when N3.4 in five consecutive overlapping 3-month periods is at or above (below) the +0.5 °C (−0.5 °C) anomaly. The El Niño (La Niña) developing year is defined as

the year (January−December) that El Niño (La Niña) develops from weak to strong, and the El Niño (La Niña) decaying year is defined when El Niño (La Niña) decays from strong to weak. Above-moderate El Niño (La Niña) events (including moderate events) are defined as when five consecutive overlapping 3-month periods are at or above (below) the 1 °C (−1 °C) SST anomaly and the largest N3.4 index value occurs from October to January. Since 1970, 9 El Niño events (1972−1973, 1982−1983, 1986−1987, 1991−1992, 1994−1995, 1997−1998, 2002−2003, 2009−2010 and 2015−2016) and 10 La Niña events (1970−1971, 1973−1974, 1975−1976, 1988−1989, 1995−1996, 1998−1999, 1999−2000, 2007−2008, 2010−2011 and 2011−2012) meet the above criteria, and in this work, unless otherwise specified, El Niño developing years are the developing years of these nine El Niño events. To increase the reliability of our results, monthly composite and regression analyses are employed in this study. The Walker index = $U_{200} − U_{850}$, where $U_{200}$ and $U_{850}$ are the mean (5°S−5°N, 120°W−120°E) zonal wind anomalies at the 200- and 850-hPa levels, respectively.

**Accumulated cyclone information calculation**. The accumulated cyclone energy (ACE)[32] in each 2° latitude × 2° longitude grid cell is defined as the sum of the squares of the estimated 6-hourly maximum sustained surface wind speed (in m s$^{-1}$) for all TCs occurring in each grid cell over all 6-h periods; i.e. the grid cell

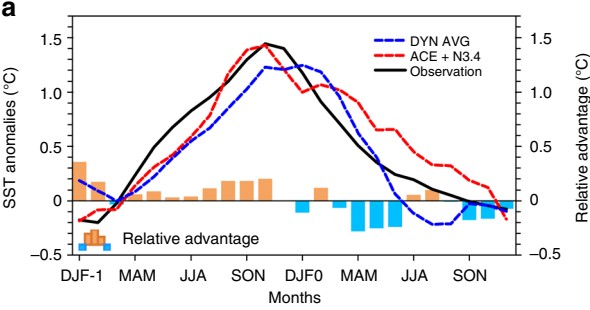

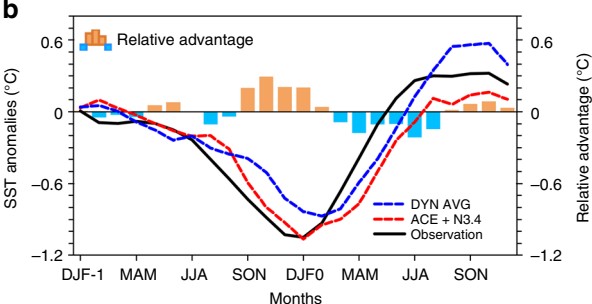

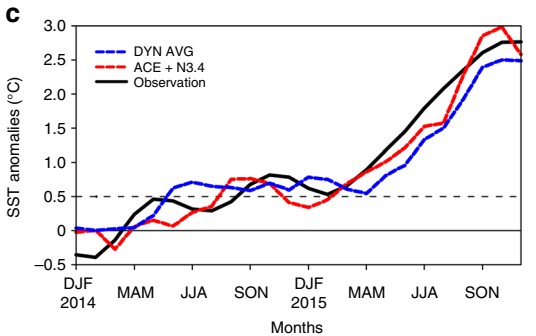

**Fig. 9** Time series of the running 3-month mean SST anomaly for the Niño 3.4 region (5°N−5°S, 120°−170°W) (N3.4, an ENSO index, °C) for observations and predictions. **a** Composite time series of N3.4 for all El Niño events in the hindcasting period (2002–2016). The blue dashed line is the average prediction by dynamical models, and the red line the ACE + N3.4 model. The black solid line is the observations. Bars indicate the amplitude of the models' relative advantage between the average of the dynamical models and the ACE + N3.4 model. Blue (orange) bars represent the advantage of the dynamical models average relative to the ACE + N3.4 model (ACE + N3.4 model relative to dynamical models average). DJF-1 and DJF0 represent the December−February in last year and the year concurring with El Niño, respectively. **b** Same as **a**, but for La Niña. **c** Time series (2014–2015) of average N3.4 from dynamical models (blue dashed line), the ACE + N3.4 model (red dashed line), and observations (black solid line)

$ACE = \sum_i V_i^2$, where $i$ is the $i$th TC in one grid cell and $V$ is its maximum sustained surface wind speed. Then, the ACE index is the anomaly of the sum of the ACE for all grid cells in the selected region. Strong (weak) ACE events are defined as having ACE index values of ≥0.5 (≤−0.5) standard deviation. A threshold value of daily ACE index in the selected region equal to 1 m²s⁻² is used to define the genesis or disappearance of TCs. A single ACE event is associated with the whole process from ACE generation to disappearance.

**Model verification**. An intermediate-complexity coupled ocean–atmosphere model[11,33] is employed to verify the role of each factor (e.g. the preceding, 3-months earlier, N3.4 and the WNP ACE) in Fig. 1b (Supplementary Fig. 4f−j). First, the horizontal wind at 925 hPa, as an initial forcing, is added to simulate the Niño-3.4 SST. The horizontal wind is obtained by simultaneous regression with respect to the corresponding factor. Then, the discrepancy between the observed and simulated SST anomalies is corrected. Finally, the simulated SST anomalies

after correction are employed to predict SST anomalies 3-months later in the model. Using TCs as an example, we first obtain the 925-hPa horizontal wind related to TCs by simultaneously regressing ACE on wind. We obtain the simulated SST anomalies by taking the 925-hPa horizontal wind related to TCs as an initial forcing. Then, the discrepancy between the observed and simulated SST anomalies is corrected. Finally, the simulated SST anomalies after correcting for errors are employed to predict SST anomalies 3 months later in the model.

**Explained percentage and linearly independent component**. Given two variables $x$ and $z$, a linear regression is calculated as follows:

$$\tilde{z}_x = ax + b \tag{1}$$

where $a$ and $b$ are the regression coefficient and a constant, respectively, and $\tilde{z}_x$ is the part of $z$ associated with $x$. The explained percentage ($p_x$) of $x$ for $z$ is

$$p_x = \frac{\tilde{z}_x}{z} \times 100\% \tag{2}$$

Let $z_x^*$ be the component of $z$ linearly independent of $x$ (also referred to as $x$-independent $z$), which can be obtained as follows:

$$z_x^* = z - \tilde{z}_x \tag{3}$$

Here, $z_x^*$ is the remainder of $z$ after removal of the signal $x$. Similarly, the explained percentage of $x$ and $y$-independent $z$ can be calculated. By the binary regression of $x$ and $y$ on $z$, $z$ associated with $x$ and $y$ ($\tilde{z}_{x,y}$) is obtained as follows:

$$\tilde{z}_{x,y} = a_1 x + a_2 y + b \tag{4}$$

where $a_1$ and $a_2$ are the regression coefficients. Then, substituting $\tilde{z}_{x,y}$ into Eq. 2, the joint explained percentage of $x$ and $y$ for $z$ is obtained.

**Lead-lag correlation**. The statistical significance of the correlation between two auto-correlated time series is assessed via a two-tailed Student's $t$-test using the effective number of degrees of freedom ($N^{eff}$). $N^{eff}$ is given by the following approximation[47–51]:

$$\frac{1}{N^{eff}} \approx \frac{1}{N} + \frac{2}{N}\sum_{j=1}^{N}\frac{N-j}{N}\rho_{XX}(j)\rho_{YY}(j) \tag{5}$$

Where $N$ is the sample size and $\rho_{XX}(j)$ and $\rho_{YY}(j)$ are the autocorrelations of the sampled time series $X$ and $Y$ at time lag $j$, respectively.

**Composite of a single TC in the period 1970-2016**. Figure 3c shows the composite time series of zonal wind anomalies (m s⁻¹) on the southern flank (within 5° of $lon_0$, and within 10° south of $lat_0$) of single TC genesis in the period 1970–2016, where $lon_0$ and $lat_0$ indicate the longitude and latitude of the TC genesis position, respectively.

**Madden–Julian Oscillation (MJO)**. A seasonally independent multivariate MJO index developed by Wheeler and Hendon (WH04)[52] is employed. This index is based on a pair of empirical orthogonal functions (EOFs) of OLR. The two leading principal components (PCs) are used to determine the daily MJO index and phase. Time series of 91-day running mean $PC1^2 + PC2^2$ defines the MJO index. In this work, a $\sqrt{PC1^2 + PC2^2} \geq 1$ defines MJO events, with others considered to be non-MJO events. A MJO is divided into eight phases. Generally, TCs in the WNP are readily generated in phases 4−7, as the convective centre propagates into the Pacific. In contrast, TC genesis is suppressed in phases 8 and 1−3 when the convective centre is located in the Indian Ocean[53]. As a result, MJO events are divided into active (phases 4−7) and inactive (phases 8 and 1−3) MJO events. The hit rate is the ratio of days in which the anomalous ACE corresponds to the MJO event type (i.e. positive ACE during active MJO events and negative ACE during inactive MJO events).

**Pre-processing of model data**. The period from 1970.01 to 1999.12 is defined as the base period. The anomalies of N3.4 and ACE in the model are calculated relative to the base period. Standardized anomalies are then calculated on the basis of the length of the chosen period. All ENSO events (no restrictions on intensity) during the hindcasting period are selected for analysis.

**Modelling methods**. Cross validation[54] is employed in the modelling process.

Holdout Method: Holdout (simple) validation[55] relies on a single partitioning of the data. The whole time series is divided into two parts: the period from January 1970 to December 1999 is the training period and the period from January 2000 to December 2016 is the hindcasting period.

With a single predictor variable, the linear regression of the predictor variable $x$ three months earlier on the dependent variable $z$ during the training period is used to construct the model. We obtain the regression coefficient $c$ and constant $g$. Thus, the forecast model can be written as

$$z_i = f_i(x) = cx_{i-3} + g \tag{6}$$

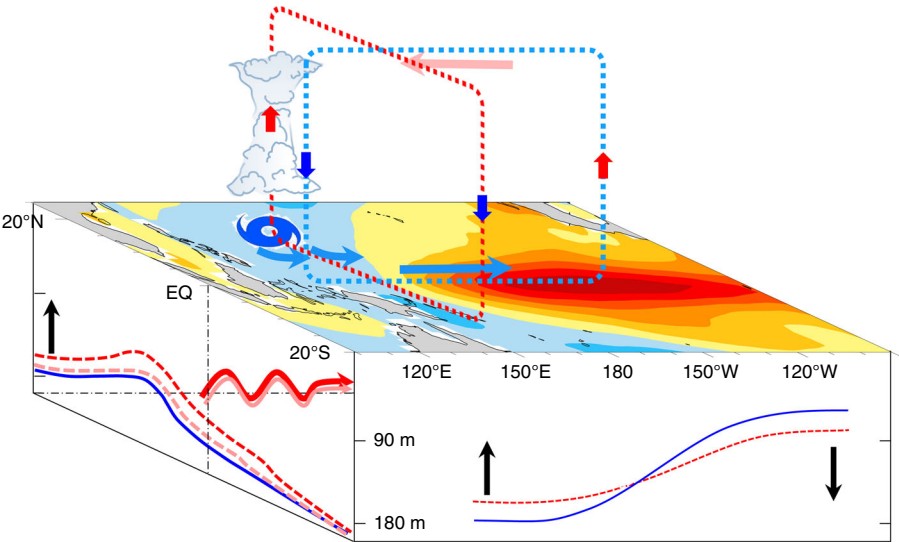

**Fig. 10** Schematic diagram of the modulation of running 3-month mean SST intensity for the Niño 3.4 region (5°N−5°S, 120°−170°W) by tropical cyclones (TCs) over the western North Pacific. Walker circulation (light blue circle) is weakened by the direct effect of asymmetrically anomalous westerlies within 0°−10°N (light blue thick arrows) related with TCs (TC symbol) at lower tropospheric levels and by the indirect effect of the Hadley-like circulation (red circle) over the tropical western Pacific. Red (blue) heavy arrows indicate updrafts (downdrafts). Moreover, TCs can shallow the thermocline (red dashed curve) in the tropical western Pacific (pink curve indicates the thermocline without TCs and blue solid line the climatological thermocline). Enhanced eastward-propagating equatorial Kelvin waves (red wavy arrow, pink wavy arrow indicates the Kelvin wave without TCs) carries warm water eastward, further deepening the thermocline in the tropical eastern Pacific, thereby reducing the gradient of the zonal thermocline in the equatorial Pacific Ocean. Both effects intensify (weaken) the El Niño (La Niña). Black arrows represent the changing direction of the thermocline

where is $f_i(x)$ a function fitted to the predictor variable $x$. This gives us the relation between the dependent variable $z$ at the $i^{th}$ month and the predictor variable $x$ at $(i-3)^{th}$ month $(x_{i-3})$ during the hindcasting period.

With two predictor variables, binary regression of the predictor variables $x$ and $y$ 3 months earlier on the dependent variable $z$ during the training period is used to construct the model. We obtain the regression coefficients $c$ and $d$, and the constant $g$. Thus, the forecast model can be represented as

$$z_i = f_i(x, y) = cx_{i-3} + dy_{i-3} + g \qquad (7)$$

where $f_i(x, y)$ is a function fitted to the predictor variables $x$ and $y$. This gives us the dependent variable $z$ at the $i^{th}$ month related to predictor variables $x$ and $y$ at the $(i-3)^{th}$ month $(x_{i-3}$ and $y_{i-3})$ during the hindcasting period.

The real predictor variables used here are the standardized ACE anomalies (ACE*) and standardized N3.4 (N3.4*) three months earlier, and the dependent variable is N3.4. Because the key domain of the ACE three months before for N3.4 varies with the type of ENSO event (key domain is 10°−20°N, 135°−170°E during an El Niño event, 10°−25°N, 130°−155°E during a La Niña event, and a common domain of 10°−20°N, 135°−155°E during other periods), three models are constructed, one for each of the three areas, according to the 0.5°C anomaly of the preceding N3.4:

$$\begin{cases} N3.4_i = c_1 ACE^*_{i-3} + g_1, N3.4_{i-3} \geq 0.5, ACE(10°-20°N, 135°-170°E) \\ N3.4_i = c_2 ACE^*_{i-3} + g_2, |N3.4_{i-3}|<0.5, ACE(10°-20°N, 135°-155°E) \\ N3.4_i = c_3 ACE^*_{i-3} + g_3, N3.4_{i-3} \leq -0.5, ACE(10°-25°N, 130°-155°E) \end{cases},$$
$$\qquad (8)$$

where $N3.4_i$ is N3.4 at the $i^{th}$ month, $ACE^*_{i-3}$ the standardized ACE anomalies at the $(i-3)^{th}$ month, and $c_1$, $c_2$, $c_3$ and $g_1$, $g_2$, $g_3$ are the regression coefficients and constants. Similarly, three ACE + N3.4 models are built, as follows:

$$\begin{cases} N3.4_i = c_4 ACE^*_{i-3} + d_1 N3.4^*_{i-3} + g_4, N3.4_{i-3} \geq 0.5, ACE(10°-20°N, 135°-170°E) \\ N3.4_i = c_5 ACE^*_{i-3} + d_2 N3.4^*_{i-3} + g_5, |N3.4_{i-3}|<0.5, ACE(10°-20°N, 135°-155°E) \\ N3.4_i = c_6 ACE^*_{i-3} + d_3 N3.4^*_{i-3} + g_6, N3.4_{i-3} \leq -0.5, ACE(10°-25°N, 130°-155°E) \end{cases}$$
$$\qquad (9)$$

Where $N3.4^*_{i-3}$ is the standardized N3.4 at the $(i-3)^{th}$ month, and $c_4$, $c_5$, $c_6$, $d_1$, $d_2$, $d_3$ and $g_4$, $g_5$, $g_6$ are the regression coefficients and constants.

Correction to the ACE + N3.4 model: Analysis shows that the role of ACE is reduced in the ACE + N3.4 model owing to the persistence of SST, especially in the development stage of ENSO. To reduce this error, the ACE + N3.4 model is corrected using generalized cross-validation (GCV)[56]. In this case, leave-one-out cross-validation (LOO-CV)[57,58] is employed. A brief description of the methodology is as follows.

Step 1. For a given time series with length $L$, one time point is chosen to be a hindcasting point, and the rest of the time series (length: $L-1$) is used to construct the forecast model with regression analysis.

Step 2. Repeat step 1; i.e. choose another time point to be a new hindcasting point, resample the residuals, and obtain a new forecast model. Repeat this process for each time point of the given time series. This will result in the ensemble hindcast.

This method will provide a new ensemble hindcast of N3.4 using the predictor variables $ACE^*$ and $N3.4^*$ The new ensemble hindcast of N3.4 produced by LOO-CV is mainly affected by the persistence of SST. Hence, by comparing this ensemble hindcast with the observations during the training period, we can obtain the correction $\varepsilon_1$ that will be used to correct the ACE + N3.4 models. And in this study, $\varepsilon_1$ depends on the base period. Because the effect of TCs on N3.4 varies with ENSO development, the whole N3.4 series is divided into four cases to obtain a more exact forecast: Case A, $N3.4_{i-3} \geq N3.4_{i-4}$ and $N3.4_{i-3} \geq 0$; Case B, $N3.4_{i-3} \geq N3.4_{i-4}$ and $N3.4_{i-3} < 0$; Case C, $N3.4_{i-3} < N3.4_{i-4}$ and $N3.4_{i-3} \geq 0$; Case D, $N3.4_{i-3} < N3.4_{i-4}$ and $N3.4_{i-3} \geq 0$. Then, four different corrected values $(\varepsilon_2)$ are obtained for the four cases during the training period. Thus, the ACE +N3.4 model is built as follows:

$$\begin{cases} N3.4_i = c_4 ACE^*_{i-3} + d_1 N3.4^*_{i-3} + g_4 + \varepsilon, N3.4_{i-3} \geq 0.5, ACE(10°-20°N, 135°-170°E) \\ N3.4_i = c_5 ACE^*_{i-3} + d_2 N3.4^*_{i-3} + g_5 + \varepsilon, |N3.4_{i-3}|<0.5, ACE(10°-20°N, 135°-155°E) \\ N3.4_i = c_6 ACE^*_{i-3} + d_3 N3.4^*_{i-3} + g_6 + \varepsilon, N3.4_{i-3} \leq -0.5, ACE(10°-25°N, 130°-155°E) \end{cases}$$
$$\qquad (10)$$

where $\varepsilon$ depends on the case, $\varepsilon = \varepsilon_1 + \varepsilon_2$.

Running holdout method: This method only differs from the holdout method in its use of a variable-length training period that changes with the hindcasting time point. That is, the required set of regression coefficients and constants is based on the time series before each hindcasting target point. The base period is the shortest training period. Other processes are as described for the holdout method.

Leave-p-out Cross-Validation (LPO-CV): Leave-p-out is similar to LOO-CV, but the single time point is replaced by a time interval. A brief description of the methodology is as follows.

Step 1. A given time series with length $L$ is divided into time intervals of length $m$. If $L$ is not exactly divisible by $m$, we suppose that the whole time series can be divided into $n$ parts and some $(<m)$ remainder time points that cannot be used as the hindcasting object.

Step 2. One part is chosen to be a hindcasting object, and the rest of the time series (length: $L-m$) is used to construct a forecast model according to regression analysis.

Step 3. Repeat step 2, using another part as a new hindcasting object, resample the residuals, and obtain a new forecast model. Repeat this process $n$ times. This will result in the ensemble hindcast without remainders.

Step 4. Compare the ensemble hindcast with the corresponding observations, and obtain correlation coefficients and root mean square errors. Then, change the value of $m$, repeat the above steps, and obtain a new ensemble hindcast.

To facilitate modelling, the start of the first time interval is 1970.04.

**Lead time**. In the model comparison, the lead time is defined by the number of months difference between the latest available observed data and the middle of the running 3-month hindcasting target period. For example, if the latest available observed data are through January, a prediction for the January–March season has a lead time of 1 month, and for February–April a lead of 2 months. Hence, in our research we mainly compare model hindcasting results at a lead of 2 months. Typically, new predictions become available one to two weeks following the last available month of observed data[14].

**Other concepts**. Sign consistency: ratio of the number of months in which the anomaly sign is predicted correctly to the total number of months.

Model's relative advantage:

$$\text{relative advantage}_{\text{model A}\rightarrow\text{model B}} = |\text{Model A} - \text{Observation}| - |\text{Model B} - \text{Observation}|$$

## Data availability

Maximum sustained surface winds and locations of TCs in the period 1970–2016 over the WNP are obtained from the International Best Track Archive for Climate Stewardship (IBTrACS; available online at https://www.ncdc.noaa.gov/ibtracs/) from the National Oceanic and Atmospheric Administration (NOAA). Monthly and daily mean wind fields for the period 1970–2016 from the National Centers for Environmental Prediction–National Center for Atmospheric Research (NCEP–NCAR) reanalysis dataset[59], on 2.5° × 2.5° global grids, are also used (available online at https://www.esrl.noaa.gov/psd/data/gridded/data.ncep.reanalysis.derived.html). Monthly SST data from 1970 to 2016 with a horizontal resolution of 2° × 2° are from the Extended Reconstructed Sea Surface Temperature (ERSST) V5 dataset[60] (available online at https://www.esrl.noaa.gov/psd/data/gridded/data.noaa.ersst.v5.html), with a horizontal resolution of 2° × 2°. The interpolated outgoing longwave radiation (OLR) data from NOAA for the period 1979–2016 are employed (available online at https://www.esrl.noaa.gov/psd/data/gridded/data.interp_OLR.html). The Simple Ocean Data Assimilation product (SODA 2.2.4) for the period 1970–2010 with 0.5° × 0.5° horizontal resolution is also employed[61] (available online at http://apdrc.soest.hawaii.edu/datadoc/soda_2.2.4.php). An ocean dataset (BOA_Argo[62]) from the China Argo Real-time Data Center is employed (available online at http://www.argo.org.cn/index.php?m=content&c=index&f=lists&catid=32). The multi-model ENSO forecast data for the period 2002–2016 are from the International Research Institute for Climate and Society provided by Columbia University (IRI, available online at http://iri.columbia.edu/~forecast/ensofcst/Data/).

## Code availability

Computer code used for the analysis was written in NCL, all types of figures that occur in this study can be found in NCL application examples (available online at https://www.ncl.ucar.edu/Applications/). More specific codes in this study are available to readers upon request.

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

## Acknowledgements

The authors are deeply indebted to Yuehong Wang, Zhaolu Hou, Jiaqing Xue, Yazhou Zhang and Quanjia Zhong for their helpful comments and suggestions. This work was jointly supported by the National Natural Science Foundation of China (NSFC, 41530424) project and State Oceanic Administration (SOA) International Cooperation Program on Global Change and Air–Sea Interactions (GASI-IPOVAI-03). C.Z.W. acknowledges the support from NSFC (41731173), the Leading Talents of Guangdong Province Program, the Pioneer Hundred Talents Program of the Chinese Academy of Sciences, the Strategic Priority Research Program of the Chinese Academy of Sciences (XDA20060502) and the National Program on Global Change and Air-Sea Interaction (GASI-IPOVAI-04).

## Author contributions

J.P.L. and Q.Y.W. contributed equally to the original idea and writing in this paper. Q.Y.W., J.P.L., F.-F.J., J.C.L.C. and C.Z.W. discussed further analysis and interpreted the results. Q.Y.W., J.P.L., F.-F.J., J.C.L.C., C.Z.W., R.Q.D., C.S., F.Z., J.F., F.X., Y.J.L., F.L. and Y.D.X. contributed to improve the manuscript.

## Additional information

**Competing interests:** The authors declare no competing interests.

