## [Peer Review File · Nature Communications]

Editorial Note: Parts of this Peer Review File have been redacted as indicated to maintain the confidentiality of unpublished data and to remove third-party material where no permission to publish could be obtained.

Reviewers' comments:

Reviewer #1 (Remarks to the Author):

Review of:

Tropical cyclones act to intensify El Niño

By:

Q. Wang et al.

Submitted to:

The authors use observations and an intermediate complexity model to examine how enhanced TC activity in the western Pacific can lead to El Niño onset by 2-3 months, thus supporting the influence of TCs on El Niño through enhanced convection, zonal wind anomalies, and Kelvin wave adjustment. The influence of TCs on El Niño is an interesting idea that has been put forth in the past in various modeling studies (Fedorov et al., 2010; Srivler et al., 2013; Li and Srivler, 2016).
<https://www.nature.com/articles/nature08831>
<https://www.earth-syst-dynam.net/4/1/2013/>
<https://agupubs.onlinelibrary.wiley.com/doi/abs/10.1002/2016JC011951>

The authors provide substantial new evidence of these links using new observational data over several decades. The paper is well-written, interesting, and very-relevant to understanding the relationship between extreme weather and larger-scale climate variability. However, the analysis lacks sufficient depth and doesn't add much to the discussion as currently presented. For example, there is little mention of the Madden-Julian Oscillation, but this could be a key factor linking TCs and El Niño since MJO can create favorable conditions for TC development and also enhanced westerly wind bursts and ocean Kelvin wave activity. Have the authors controlled for MJO in the analysis of weak versus active conditions? [REDACTED]

Providing insights to these deeper questions would improve the paper considerably. I also suggest providing more discussion about the importance of these results and broader impacts. For example, the closing paragraph states that "results show the climatic effect of TC activity should be given more attention, which would lead to improved understanding of ENSO and other climatic events." This is vague, and it's not clear how the results support this. To what

extent are El Niño events enhanced? Would they not occur if TCs didn't happen? Would they be weaker?

This is a potentially very strong paper and good fit for Nature Communications with revisions. I suggest the authors dig a bit deeper in the analysis (e.g. MJO) to extend on the earlier model-based studies, and improve the communications part of the paper (why is this important in the broader context of climate change?)

Reviewer #2 (Remarks to the Author):

The basis of this paper is interesting... A measure of tropical cyclone activity in the southeastern part of the western North Pacific is strongly correlated with an El Niño index 3 months later. This is not exactly a new observation, having been made by Camargo and Sobel in 2005 (as the authors state), but those authors attributed most of the lagged signal to auto-correlation of sea surface temperature (SST). The current authors attempt to make the case that there is a causal link from the tropical cyclones to the increase in SST further east, 3 months later. This is plausible since TCs are associated with strong westerlies on the equator. (Although the authors do not mention it, it is not entirely clear that such westerlies are actually "caused" by the TCs...there is a literature suggesting that there are westerly wind bursts on the equator that precede TC formation cause the TCs.) Also, Fedorov et al. suggest an oceanic route for TC influences on El Niño that is not quite the same as what the current authors describe. On the other hand, this reviewer is not persuaded

that there is causal link from the TCs to El Niño. [REDACTED]

I would have to be persuaded that this simple hypothesis is wrong before wanting to pursue the more complex hypothesis of a causal chain from the TCs to El Niño.

The section on causality is far too speculative for a high profile paper in Nature Communications. It would be better if the authors could test their hypothesis using a coupled model.

I. Response to Comments of Reviewer #1

Reviewer #1

The authors use observations and an intermediate complexity model to examine how enhanced TC activity in the western pacific can lead El Nino onset by 2-3 months, thus supporting the influence of TCs on El Nino through enhanced convection, zonal wind anomalies, and kelvin wave adjustment. The influence of TCs on El Nino is an interesting idea that has been put forth in the past in various modeling studies (Fedorov et al., 2010; Sriviver et al, 2013; Li and Sriviver, 2016).

<https://www.nature.com/articles/nature08831>

<https://www.earth-syst-dynam.net/4/1/2013/>

<https://agupubs.onlinelibrary.wiley.com/doi/abs/10.1002/2016JC011951>

The authors provide substantial new evidence of these links using new observational data over several decades. The paper is well-written, interesting, and very-relevant to understanding the relationship between extreme weather and larger-scale climate variability. However, the analysis lacks sufficient depth and doesn't add much to the discussion as currently presented. For example, there is little mention of the Madden Julian Oscillation, but this could be a key factor linking TCs and El Nino since MJO can create favorable conditions for TC development and also enhanced westerly wind bursts and ocean kelvin wave activity. Have the authors controlled for MJO in the analysis of weak versus active conditions? [REDACTED] Providing insights to these deeper questions would improve the paper considerably. I also suggest providing more discussion about the importance of these results and broader impacts. For example, the closing paragraph states that "results show the climatic effect of TC activity should be given more attention, which would lead improved understanding of ENSO and other climatic events." This is vague, and it's not clear how the results support this. To what extent are El Nino events enhanced? Would they not occur if TCs didn't happen? Would they be weaker?

This is a potentially very strong paper and good fit for Nature Communications with revisions. I suggest the authors dig a bit deeper in the analysis (e.g. mjo) to extend on the earlier model-based studies, and improve the communications part of the paper (why is this important in the broader context of climate change?)

Response:

Firstly, thank the reviewer’s support for our study and the constructive comments, which give us a big help to improve the quality of our manuscript. Secondly, in view of the reviewer’s comments and suggestions, we gave point-by-point response and have added precisely and revised carefully in this manuscript as follows.

Q1 “The influence of TCs on El Nino is an interesting idea that has been put forth in the past in various modeling studies (Fedorov et al., 2010; Sriver et al, 2013; Li and Sriver, 2016).”

Response:

Thanks reviewers’ reminder. We must admit that the studies the reviewer mentioned give us a big help to strengthen the paper and enlighten us to show further robust evidence and physical mechanism which provide a solid basis for our new findings. However, it may cause some misunderstanding to the originality of our work, hence, detailed comparisons between our and previous studies have been shown in Tables A1 and A2.

Table A1 Comparison between our and Fedorov et al.’s work (2010)

	Fedorov et al.’s work (2010)	Our study
Timescales	They described “a positive feedback between hurricanes and the upper-ocean circulation in the tropical Pacific Ocean that may have been essential for maintaining warm, El Niño-like conditions during the early Pliocene ”	We study the modulation of TCs over western Pacific to El Niño on interannual timescale in modern times, which is quite another thing.
Physical mechanism	They thought “ In the present climate, very few hurricane tracks intersect the parcel trajectories ”, which implies TCs will not play a main role in El Niño event in modern times.	We think TCs play an important role in intensity of El Niño event in modern times and propose an entirely different mechanism.
TCs’ distribution	For the early Pliocene, TCs cover two bands of the entire zonal extent of the Pacific	In modern time, there is no TCs in the central Pacific.

Table A2 Comparison between our and Sriver et al.’s work (2013 & 2016)

	Sriver et al.’s work (2013 & 2016)	Our study
Key finding	In their 2013 work , they study was based on a special case of 3 successive tropical cyclones occurring in the western North Pacific (during May 2003) in an ocean general circulation model. And by blending into global atmospheric forcing from 1997, and the control simulation reproduced the main characteristics of the observed zonal redistribution of ocean heat in the tropical Pacific during 1997. Their work was a resolution-dependent special case analysis. Also in their 2016 work , they said “A key finding of this study is that tropical cyclones could significantly contribute to global ocean heat and energy budgets, and the magnitude of which depends on ocean grid resolution. ”	As the reviewer said “ The authors provide substantial new evidence of these links using new observational data over several decades ”, we reveal the universal and substantial effect of TCs on El Niño events.
Relationship between ACE and El Niño on interannual timescale	They didn’t mention it.	We firstly report that the TC activity in the southeastern western North Pacific (WNP) leads El Niño by about 3 months from the perspective of the accumulated cyclone energy (ACE) index.
Mechanism	They just mentioned excitation of equatorial Kelvin and Yanai waves by TCs can contribute to global ocean heat and energy budgets.	We show atmospheric and oceanic process to analyze the modulation of TCs to El Niño. It’s undeniable that their result can help us to better understand the oceanic process in our study, but it’s not entire route that TCs effect El Niño in oceanic process.

Hence, the originality of our work is strong enough, and these results mentioned above can give some support to our results and boost our confidence in our results. And these study have been mentioned in the revised manuscript (lines:47, 113,160, 215).

Q2 “the analysis lacks sufficient depth and doesn’t add much to the discussion as currently presented... Have the authors controlled for MJO in the analysis of weak versus active conditions?”

Response:

- **The effect of MJO on modulation of TCs to ENSO intensity.**

Previous studies (McPhaden 1999, Puy et al. 2016) indicate MJO may play an important role on ENSO, especially onset of El Niño and also effecting TC activities over the north Western Pacific in intraseasonal timescale. Actually this work doesn't involve the onset of El Niño, and we also appreciate the reviewer gives us a valuable reminder to further verify our result.

1) Data and methods

The interpolated outgoing longwave radiation (OLR) from National Oceanic and Atmospheric Administration (NOAA) in the period 1979–2016 is employed. Daily sea surface temperature (SST) in the period 1979–2016 is obtained from ERA Interim dataset of European Centre for Medium-Range Weather Forecasts (ECMWF).

Table A3 The criterion about MJO and non-MJO events

events	criterion
non-MJO	$\sqrt{PC_1^2 + PC_2^2} < 1$
MJO_active	$\sqrt{PC_1^2 + PC_2^2} \geq 1$ and Phase 4-7
MJO_inactive	$\sqrt{PC_1^2 + PC_2^2} \geq 1$ and Phase 8,1-3

A seasonally independent multivariate MJO index that Wheeler and Hendon developed (WH04) (Wheeler and Hendon 2004) is employed. This index is based on a pair of empirical orthogonal functions (EOFs) of OLR. The two leading principal components (PCs) are used to determine the daily MJO index and phase. As shown in Table A3, time series of 91-day running mean $PC1^2 + PC2^2$ defines the MJO index, in this work, the $\sqrt{PC1^2 + PC2^2}$ greater than or equal to 1 defines MJO event, and others are non-MJO events. MJO is divided into eight phases. Generally, TCs over WNP are easily to generate in phase 4–7 while the convective center propagates into the Pacific. On the contrary, TC genesis is suppressed in phased 8 and 1–3 while the convective center is located in the Indian Ocean (Zhao et al. 2015). As a result, MJO events are divide into active (phase 4–7) and inactive (phase 8 and 1–3) MJO events. After removing MJO signal from the time series of daily SST anomalies, and converting daily

data to monthly data, we can obtain the new series of monthly SST anomalies without MJO. Then a 3-month running mean is taken to new series. Monthly accumulated cyclone energy index (ACE) anomalies after removing MJO signal can also be obtain by same method.

Fig. A1 Lead-lag correlations between WNP ACE index and N3.4 and their autocorrelations in the period 1979–2016. **a**, Lead-lag correlation and autocorrelations between original series. The red, green and blue dashed lines indicate significant at the 99% confidence level related to lead-lag correlations between WNP ACE index and N3.4, autocorrelations of N3.4 and WNP ACE index via Student’s *t*-test using the effective number of degrees of freedom, respectively. **b**, Lead-lag correlations between the processed series. $N3.4_{N3.4_0^*}$ ($ACE_{N3.4_0^*}$) indicates the N3.4 (WNP ACE index) that is not associated with the preceding (three months earlier) N3.4. $ACE_{N3.4^*}$ indicates the WNP ACE index that is not associated with the simultaneous N3.4. The red, green and blue dashed lines indicate significant at the 99% confidence level related to lead-lag correlations between N3.4 and $ACE_{N3.4_0^*}$, N3.4 and $ACE_{N3.4^*}$, $N3.4_{N3.4_0^*}$ and $ACE_{N3.4_0^*}$ via Student’s *t*-test using the effective number of degrees of freedom, respectively. **c**, Same as **a**, but for series after removing MJO. **d**, same as **b**, but for series after removing MJO

2) The MJO signal does not affect the modulation of TCs to El Niño intensity on

interannual timescale.

Fig. A1 shows the lead-lag correlation between the monthly N3.4 and ACE. Firstly, when the SST data from the ERA Interim dataset is employed [in previous manuscript, monthly SST is from the Extended Reconstructed Sea Surface Temperature (ERSST) V5 dataset of NOAA], the result that ACE in the southeastern western North Pacific (WNP) is strongly correlated with Niño 3.4 (N3.4) three months later is robust (Figs A1a and b). Secondly, as can be seen from Figs A1c and d, after removing MJO signal, this relationship between ACE and SST is hardly affected, and the amplitude of correlation coefficient changes a slight. That is, this MJO signal does not affect the modulation of TCs to ENSO intensity on interannual timescale.

3) The weak or active conditions of MJO do not play different role during the modulating process of TCs to El Niño intensity

In this study, we find the WNP TCs in July–September (J-A-S) can significantly intensify El Niño in October–December (O-N-D); and the greater the ACE, the stronger El Niño stronger. Hence, we explore the relationship between daily TC activities and MJO phases in J-A-S.

Fig. A2 shows the phase distribution of daily MJO index in J-A-S during eight El Niño developing years. Active and inactive MJO events as well as non-MJO events occur in J-A-S during eight El Niño developing years. The difference among the occurrence proportions of three events is nonsignificant, and the non-MJO events seem most often, especially in July (Fig. A3). This result implies that the modulation of TCs to El Niño intensity on interannual time scale may be independent of the MJO situation. In order to further verify this, we explore the hit rates, the ratios of days in which the anomaly ACE is correctly to the situation of MJO event (that is, positive ACE is for active MJO event and negative ACE for inactive MJO event). From Table A4, hit rates hold low regardless of single years or composite year, except for 2015, which means MJO does not mainly effect to during the modulating process of TCs to El Niño intensity. [REDACTED]

About the role of MJO in the modulating of TCs to El Niño intensity has been added in the revised manuscript at lines: 186-200.

Fig. A2 Phase distribution of daily MJO index in J-A-S during eight El Niño developing years in the period 1979–2016. **a**, July. Big dot indicates the first day of each month, small dot one day, each line one year. **b**, August. **c**, September.

Fig. A3 Occurrence proportion (%) of days relating to three events (non-MJO, active MJO and inactive MJO) in J-A-S during eight El Niño developing years in the period 1979–2016.

Table A4 Hit rates between ACE and MJO events in El Niño developing years and composite year.

	July	August	September
1982	16.13	61.29	33.33
1986	6.45	58.06	30.00
1991	6.45	38.71	20.00
1994	0.00	0.00	20.00
1997	9.68	0.00	23.33
2002	9.68	22.58	0.00
2009	35.48	58.06	56.67
2015	90.32	45.16	63.33
average	21.77	35.48	30.83

[REDACTED]

References

- McPhaden, M. J., 1999: Genesis and evolution of the 1997-98 El Nino. *Science*, **283**: 950-954.
- Puy, M., J. Vialard, M. Lengaigne and E. Guilyardi, 2016: Modulation of equatorial Pacific westerly/easterly wind events by the Madden-Julian oscillation and convectively-coupled Rossby waves. *Clim Dynam*, **46**: 2155-2178.
- Wheeler, M. C. and H. H. Hendon, 2004: An all-season real-time multivariate MJO index: Development of an index for monitoring and prediction. *Mon Weather Rev*, **132**: 1917-1932.
- Zhao, H. K., R. Yoshida and G. B. Raga, 2015: Impact of the Madden-Julian Oscillation on Western North Pacific Tropical Cyclogenesis Associated with Large-Scale Patterns. *J Appl Meteorol Clim*, **54**: 1413-1429.

Q3

[REDACTED]

Q4 “I also suggest providing more discussion about the importance of these results and broader impacts. For example, the closing paragraph states that “results show the climatic effect of TC activity should be given more attention, which would lead improved understanding of ENSO and other climatic events.” This is vague, and it’s not clear how the results support this. To what extent are El Nino events enhanced? Would they not occur if TCs didn’t happen? Would they be weaker”

Response:

Thanks for the reviewer’s warm reminder, as the reviewer’s saying, more discussion should be added in the closing paragraph indeed. For example, we give the related response about the question that “To what extent are El Nino events enhanced? Would they not occur if TCs didn’t happen? Would they be weaker”: Furthermore, by the observed explained percentage of N3.4 intensity obtained by preceding contributors’ signals during El Niño developing years (as shown in Fig. 1), it can be estimated that the modulated extent of WNP TCs to El Niño intensity will reach about 20%. The absence of TCs would weaken the intensity of El Niño.” (at lines: 219-221 in the revised manuscript). And more detailed have been added in section results and discussions, as follows:

“The climatic influence of WNP TCs on El Niño is examined in this work. Our main result is that WNP TCs can significantly intensify El Niño three months later, especially for J-A-S TCs. The greater the WNP ACE, the stronger the El Niño. And this modulation is independent of MJO. WNP TCs in J-A-S can affect both the atmospheric and oceanic processes that drive the O-N-D El Niño (Fig. 8 and Extended Data Fig. 10). WNP TCs can weaken the Walker circulation via direct effect of asymmetrically anomalous westerlies at lower tropospheric level and indirect effect of Hadley-like circulation. TCs can shallow the thermocline in the tropical Western Pacific. Warm water in the tropical Western Pacific are carried eastward, associating with the enhanced eastward propagating equatorial Kelvin wave, further deepening the thermocline in the tropical eastern Pacific, thereby reducing the gradient of zonal thermocline in the equatorial Pacific. This two processes both intensify El Niño. And this detailed effect process is different from that during the early Pliocene epoch18 and short-term ACE

case (for example single TC case 25, 26, it's no doubt that they can cause the change of SST, but their short-term ACE is not enough strong to maintain interannual variability of N3.4). In fact, TCs can also affect La Niña (not shown). Furthermore, by the observed explained percentage of N3.4 intensity obtained by preceding contributors' signals during El Niño developing years (as shown in Fig. 1), we can estimate this modulated extent of WNP TCs to El Niño intensity will reach about 20%. The absence of TCs would weaken the intensity of El Niño. These results show that this neglected relationship between TCs and ENSO may lead to an improved understanding of ENSO, further helping to improve ENSO amplitude forecasts. Thus further research is needed to verify and quantize the extent of this modulation, and determine how this relationship can be incorporated into existing forecasting models. In addition, these results also remind us that the climatic effect of TC activity should be given more attention, even on other climatic events.”

II. Response to Comments of Reviewer #2

Reviewer #2

The basis of this paper is interesting... A measure of tropical cyclone activity in the southeastern part of the western North Pacific is strongly correlated with an El Niño index 3 months later. This is not exactly a new observation, having been made by Camargo and Sobel in 2005 (as the authors state), but those authors attributed most of the lagged signal to auto-correlation of sea surface temperature (SST). The current authors attempt to make the case that there is a causal link from the tropical cyclones to the increase in SST further east, 3 months later. This is plausible since TCs are associated with strong westerlies on the equator. (Although the authors do not mention it, it is not entirely clear that such westerlies are actually “caused” by the TCs...there is a literature suggesting that there are westerly wind bursts on the equator that precede TC formation cause the TCs.) Also, Federov et al. suggest an oceanic route for TC influences on El Niño that is not quite the same as what the current authors describe. On the other hand, this reviewer is not persuaded that there is causal link from the TCs to El Niño. [REDACTED] I would have to be persuaded that this simple hypothesis is wrong before wanting to pursue the more complex hypothesis of a causal chain from the TCs to El Niño.

The section on causality is far too speculative for a high profile paper in Nature Communications. It would be better if the authors could test their hypothesis using a coupled model.

Response:

Thanks to the reviewer for sparing time to go through the manuscript, highlighting very important issues and providing valuable suggestions to improve the manuscript.

The manuscript has been revised carefully and more supporting materials have been added. More details and point-to-point responses to the reviewer’s comments are listed as follows.

Q1 “A measure of tropical cyclone activity in the southeastern part of the western North Pacific is strongly correlated with an El Niño index 3 months later...but those authors attributed most of the lagged signal to auto-correlation of sea surface temperature (SST).”

Response:

- **The result that accumulated cyclone energy index (ACE) over the western North Pacific (WNP) leads Niño 3.4 index (N3.4) reaching maximum at lead three months in the interannual time scale is exactly a new observation, and this result is independent of seasons.**

We’re sorry that the inaccurate description in the manuscript may make the reviewer’s misunderstanding. Table B1 shows detailed comparison between our and Camargo and Sobel’s study (2005).

Table B1 comparison between our and Camargo and Sobel’s study (2005)

	Camargo and Sobel’s work (2005)	Our study
ACE calculation	Traditional calculated method of ACE was employed. “ACE is defined as the sum of the squares of the estimated 6-hourly maximum sustained surface wind speed for all TCs in the basin summed over all 6-h periods”. ACE depends on each TC genesis position.	ACE in each 2° latitude × 2° longitude grid cell is defined as the sum of the squares of the estimated 6-hourly maximum sustained surface wind speed for all TCs occurring in each grid cell over all 6-h periods. ACE is independent of the TC genesis position, it can better show the actual distribution of each grid cell than that using traditional calculated method.
Entirely different observation	As shown in Fig.3 in their paper, it was obviously that the maximum correlation between ACE and Niño 3.4 occurred when Niño 3.4 lead ACE.	As shown in Fig. B1a, during the same period as that in Camargo and Sobel’s work, it is obviously that the maximum correlation between ACE and Niño 3.4 occur when ACE leads N3.4. During our study’s period, this lead relationship is more significant (Fig. B1b). This entirely different observation is owing to the ACE calculation. In the whole basin, the sum value of ACE is same as that using traditional method, but to one part of basin, it’s quite different. And in our study, ACE in the southeastern western North Pacific is employed.

[REDACTED]

Fig. B1 Lag correlations of ACE over the southeastern western North Pacific (July–October, JASO) with Niño 3.4 for different seasons (months denoted by their first letter). **a**, In the period 1970–2002 (Same as the period in Camargo and Sobel’s work). Red line is the maximum correlation between ACE and N3.4. Blue box the simultaneous correlation. The red and blue dashed lines indicate significant at the 99%, 95% confidence levels, respectively. **b**, Same as **a**, but for the period 1970–2016 (Same as the period in our study).

Hence, our study differs entirely from Camargo and Sobel’s work. And this part has been mentioned in the revised manuscript (lines:57-67).

Reference

Camargo, S. J. and A. H. Sobel, 2005: Western North Pacific tropical cyclone intensity and ENSO. *J Climate*, **18**: 2996-3006.

Q2 “This is plausible since TCs are associated with strong westerlies on the equator. (Although the authors do not mention it, it is not entirely clear that such westerlies

are actually “caused” by the TCs...there is a literature suggesting that there are westerly wind bursts on the equator that precede TC formation cause the TCs.)”

Response:

- **Daily zonal wind near equator has not a main influence on interannual climatic effect of tropical cyclone (TC) activities over WNP on N3.4 indeed**

The mean values of maximum zonal wind anomalies with and without TCs occurrence are shown in Fig. B2. We can find that the westerlies wind anomalies with TCs is stronger than that without TC occurrence regardless of whether it is an El Niño developing year. And the ratio between situation with TCs and that without TCs holds approximately equal. This result implies this enhancement of TCs to westerlies wind is independent on the El Niño occurrence.

Fig. B2 Mean values of maximum zonal wind anomalies (m s^{-1}) and the ratios of mean values between maximum zonal wind anomalies with TCs and that without TCs.

The relationship between WNP ACE index after removing daily zonal wind near the equator and N3.4 is also checked. Area-averaged zonal wind anomaly near the equator ($5^{\circ}\text{S}-5^{\circ}\text{N}$, $120^{\circ}\text{E}-180^{\circ}$) defines a zonal wind index. Using partial correlation, we obtain daily WNP ACE after removing daily zonal wind near the equator, then converting daily data to monthly data. As shown in Fig. B3a, after removing the effect of daily zonal wind near the equator on WNP ACE, WNP ACE still leads N3.4 by about 3 months. The amplitude of maximum correlation coefficient between WNP ACE and N3.4 decreases, but it is still significant. And what we need mention is that the removed daily zonal wind near the equator may include the effect of daily WNP ACE. Also, from the Fig. B3b, this leaded signal still hold when we remove simultaneous and proceeding (three months earlier) SST signals. Hence, we can't deny daily zonal wind near equator may provide a favorable condition, but daily zonal wind near equator has not a main

influence on interannual climatic effect of TCs activities over the WNP on N3.4 indeed.

Fig. B3 Lead-lag correlations between WNP ACE index after removing daily zonal wind near equator and N3.4 and their autocorrelations in the period 1970–2016. a, Lead-lag correlation and autocorrelations between original series. The red, green and blue dashed lines indicate significant at the 99% confidence level related to lead-lag correlations between WNP ACE index and N3.4, autocorrelations of N3.4 and WNP ACE index via Student’s *t*-test using the effective number of degrees of freedom, respectively. **b,** Lead-lag correlations between the processed series.

N3.4_N3.4₀^{*} (ACE_N3.4₀^{*}) indicates the N3.4 (WNP ACE index) that is not associated with the preceding (three months earlier) N3.4. ACE_N3.4^{*} indicates the WNP ACE index that is not associated with the simultaneous N3.4. The red, green and blue dashed lines indicate significant at the 99% confidence level related to lead-lag correlations between N3.4 and ACE_N3.4₀^{*}, N3.4 and ACE_N3.4^{*}, N3.4_N3.4₀^{*} and ACE_N3.4₀^{*} via Student’s *t*-test using the effective number of degrees of freedom, respectively.

Q3 “Fedorov et al. suggest an oceanic route for TC influences on El Niño that is not quite the same as what the current authors describe.”

Response:

Thanks for reviewer’s question, firstly, the main comparisons between Fedorov et al.’s work (2010) and ours in following Table B2.

Table B2 comparison between our and Fedorov et al.’s work (2010)

	Fedorov et al.’s work (2010)	Our study
Timescales	They described “a positive feedback between hurricanes and the upper-ocean circulation in the tropical Pacific Ocean that may have been essential for maintaining warm, El Niño-like conditions during the early Pliocene”.	We study the modulation of TCs over western Pacific to El Niño in modern times, which is quite another thing.
Physical mechanism	They thought “In the present climate, very few hurricane tracks intersect the parcel trajectories”, which implies TCs will not play a main role in El Niño event in modern times.	We think TCs play an important role in intensity of El Niño event in modern times, and propose an entirely different mechanism.

TCS' distribution	For the early Pliocene, TCs cover two bands of the entire zonal extent of the Pacific.	In modern time, there is hardly any TCs in the central Pacific.
---	--

- **Oceanic process for TC influences on El Niño**

In order to verify TCs can change the oceanic situation and further affect the El Niño on interannual timescale, the monthly mean equatorial potential temperature anomalies are examined.

Fig. B4 Composite of depth–zonal distribution of monthly equatorial potential temperature anomalies averaged between 5°S–5°N (shading; °C) during the El Niño developing year (2004–2016). The dashed contours denote the isotherms of the potential temperature.

Fig. B4 is composite of monthly mean equatorial potential temperature anomalies during the El Niño developing year. Because of the limit of ocean data (2004–2016, BOA_Argo dataset from China Argo Real-time Data Center), we just choose 2015 and 2009 as the El Niño developing year. The positive potential temperature anomalies began to emerge in the western Pacific and then propagated eastward along the main thermocline and accumulated in the equatorial eastern Pacific. Starting from June, the negative potential temperature anomalies are growing in the western Pacific because of the eastward transportation of surface warm water and upward cold water, which further helps the enhanced positive potential temperature anomalies in the equatorial eastern Pacific by transporting surface warm water. After removing the simultaneous TCs signal (Fig. B5), this change of potential temperature anomalies decrease, even disappear, especially in WNP TC season (May–October) (Fig. B6). Also, when the proceeding (three months earlier) ACE signal is removed, the potential temperature anomalies are also decreased, even disappeared, especially in three months after TC

season (Figs B7 and B8). All of observations prove that WNP TCs can enhanced the transportation of warm water from equatorial western to eastern Pacific.(This part has added at lines 186-200 in the revised manuscript.

Fig. B5 Same as Fig. B4, but for that after removing the simultaneous WNP ACE.

Fig. B6 Same as Fig. B4, but for that relating to the simultaneous WNP ACE.

Fig. B7 Same as Fig. B4, but for that after removing the preceding (three months earlier) WNP ACE.

Fig. B8 Same as Fig. B4, but for that related to the preceding (three months earlier) WNP ACE.

Reference

Fedorov, A. V., C. M. Brierley and K. Emanuel, 2010: Tropical cyclones and permanent El Nino in the early Pliocene epoch. *Nature*, **463**: 1066-U1084.

Q4

[REDACTED]

Q5 “It would be better if the authors could test their hypothesis using a coupled model.”

Response:

Thanks for the reviewer’s good suggestion. Previous studies (Fedorov et al., 2010, Buetti et al., 2014, Li et al., 2016) have provided fundamental insights into simulating climatic TCs in a fully coupled model, but it is only restricted to TCs’ climatic distribution. For example, in Fedorov et al. study (2010), they simulated the overall distribution TC tracks in the modern and the early Pliocene climate. Li et al. (2016) showed overall storm tracks accumulated over 20 years in the different resolutions, and their study also pointed the model-simulated accumulated TCs were generally limited by relatively coarse ocean grid resolution and a lack of appropriate for analyzing TC effects in the coupled system. So far, no published study has indicated the daily or monthly activities of basin-scale TCs (not single TC events) can be simulated in current coupled models. It’s scarcely possible to find a fully coupled model that can simulate the interannual variability of both TC activities and El Niño well. Hence, we are sorry about that so far we have no way to show how WNP TCs effect El Niño on interannual timescale by the method of ocean-atmosphere coupling.

In manuscript, we show the result from an intermediate complexity coupled ocean–atmosphere model under giving atmospheric forcing. Due to our carelessness, we didn’t give detailed explanation to this part in last version of manuscript. Although it’s not enough complex rather than current fully couple models, previous studies (Cane et al. 1986, Zebiak and Cane 1987, Chen et al. 2004) have proved that it can capture well the features of El Niño. We check its simulated ability for El Niño before we use it. From Fig. B11 (Extended Data Fig. 5 in the manuscript), we can find whether the time series of N3.4 or spatial pattern of SST anomalies between observation and prediction, are very similar, the correlation coefficient reaches 0.86 between them, and the intensity of N3.4 is also approximately. The model shows similar result with the observations (Fig. B12). In models, the mean explained percentage of the N3.4-independent ACE in J-A-S to N3.4 in O-N-D is equivalent to that of the ACE-independent N3.4 (14.3% vs 16%, respectively). The joint explained percentage of the preceding ACE and N3.4 is nearly 18% higher than that of either factor alone. All these features imply ACE have important role in the intensity of El Niño. (We have added the related descriptions at lines 77-82 in the revised manuscript.)

Fig. B11 Time series of the succeeding (three months later) N3.4 (°C) from 1970 to 2016 and spatial distributions of SSTAs (shading, °C) during the El Niño developing years. **a**, Time series of the observed N3.4 (red line) and predicted (black line) N3.4 from an intermediate complexity coupled ocean–atmosphere model (see **Methods**). r is correlation coefficient; **b**, Observation; **c**, Predicted result. The black rectangle denotes the selected the Niño-3.4 region (5°S–5°N, 120°–

170°W).

Fig. B12 Explained percentage (%) of N3.4 intensity from October to December obtained by preceding (3 months earlier) contributors' signals during El Niño developing years (1970–2016). **a**, Observations. ACE₀&N_{3.4}₀ represents the combined contributor of the preceding WNP ACE index and N_{3.4}; N_{3.4}₀ (ACE₀) the preceding N_{3.4} (WNP ACE index); N_{3.4}^{*} the preceding ACE-independent N_{3.4} (see Methods); ACE₀^{*} the preceding N_{3.4}-independent ACE index (see Methods). Black (grey) bold font represents significant above the 95% (90%) confidence level using Student's t-test. **b**, Same as **a**, but for the intermediate complexity coupled ocean–atmosphere model.

References

Bueti, M. R., Ginis, I., Rothstein, L. M. & Griffies, S. M. Tropical Cyclone-Induced Thermocline Warming and Its Regional and Global Impacts. *J Climate* **27**, 6978–6999, doi:10.1175/Jcli-D-14-00152.1 (2014).

Cane, M. A., S. E. Zebiak and S. C. Dolan, 1986: Experimental forecasts of El-Niño. *Nature*, **321**: 827-832.

Chen, D., M. A. Cane, A. Kaplan, S. E. Zebiak and D. J. Huang, 2004: Predictability of El Niño over the past 148 years. *Nature*, **428**: 733-736.

Fedorov, A. V., C. M. Brierley and K. Emanuel, 2010: Tropical cyclones and permanent El Niño in the early Pliocene epoch. *Nature*, **463**: 1066-U1084.

Li, H., R. L. Sriver and M. Goes, 2016: Modeled sensitivity of the Northwestern Pacific

upper-ocean response to tropical cyclones in a fully coupled climate model with varying ocean grid resolution. *J Geophys Res-Oceans*, **121**: 586-8601.

Zebiak, S. E. and M. A. Cane, 1987: A Model El-Niño Southern Oscillation. *Mon Weather Rev*, **115**: 2262-2278.

Reviewers' comments:

Reviewer #1 (Remarks to the Author):

The manuscript is improved. The ideas are interesting and fit with recent literature (though they go into much more depth here). Conclusions are still a bit speculative, and I suggest adding in some text on caveats/limitations based on the reviewers' points. I also suggest a careful edit for English. I recommend acceptance after these minor recommendations.

Reviewer #2 (Remarks to the Author):

The authors have done a commendable amount of work to address my earlier review. At the same time, my concerns about the causal chain being argued for have not been much alleviated. For example, Figure 3b clearly shows westerly winds on the equator before the TCs develop, at least as indicated by ACE, and I see no evidence that external influences such as the MJO have been ruled out. Beyond that, my earlier complaint that there are many rather speculative statements stands. For example, on lines 128-130, it is stated that 'The main reason for this is a Hadley-like circulation (20°N–20°S, 135°–170°E) caused by the TCs'. How do we know that it was caused by the TCs?

It would be far better to try to publish this paper in a journal that specializes in ENSO-related research so that experts in the field can begin a discussion of the ideas presented here, before attempting to present it to a broader audience.

I. Response to Comments of Reviewer #1

Reviewer #1

The manuscript is improved. The ideas are interesting and fit with recent literature (though they go into much more depth here). Conclusions are still a bit speculative, and I suggest adding in some text on caveats/limitations based on the reviewers' points. I also suggest a careful edit for English. I recommend acceptance after these minor recommendations.

Response:

1. Caveats/limitations

Thanks for the reviewer's warm reminder, as the reviewer's saying, more text on caveats/limitations should be added in the closing discussion paragraph indeed. We have divided the original closing paragraph to two paragraphs: the key results in the first part and a detailed discussion including limitations in the second part.

The first paragraph (lines: 212-225): "The feedback of WNP TCs on El Niño on interannual timescale is examined using observational data over several decades in this work. It is found that WNP TCs can significantly intensify El Niño 3-months later, particularly for J-A-S TCs. The greater the WNP ACE, the stronger the El Niño. This modulation has few dependencies on the MJO. WNP TCs in J-A-S can affect both the atmospheric and oceanic processes that drive the O-N-D El Niño (Fig. 8 and Extended Data Fig. 12), which is different from that during the early Pliocene epoch¹⁸. WNP TCs can weaken the Walker circulation via direct effects of equatorial asymmetrically anomalous westerlies at lower tropospheric levels and indirect effects of Hadley-like circulation. TCs can shallow the thermocline in the tropical Western Pacific. Warm water in the tropical Western Pacific is carried eastward, in association with the enhanced eastward-propagating equatorial Kelvin waves, further deepening the thermocline in the tropical eastern Pacific, thereby reducing the gradient of the zonal thermocline in the equatorial Pacific. These two processes lead to intensified El Niño. TCs can also affect La Niña (not shown)."

The second paragraph (lines: 226-243): "Although the primary focus here is on

the feedback of WNP TC occurrence on westerlies and the thermocline, other related topics warrant further investigation. For example, the effect of the MJO and other factors on TC genesis and the possible physical mechanisms associated with the influence of WNP TCs on Hadley-like circulations and tropical westerlies still need to be verified in a fully coupled model. Previous studies^{18,26,30} have provided fundamental insights into simulating climatic TCs in a fully coupled model, but were restricted to TC climatic distributions (or to accumulated TCs). Currently, fully coupled models that perform well in simulating interannual variabilities of both TC activity and El Niño are rare, and remain an important challenge for researchers. Furthermore, the observed explained percentage of N3.4 intensity during the El Niño developing years in this work (Fig. 1) shows that the modulated strength of WNP TCs to El Niño intensity may reach ~20%. The absence of TCs can weaken El Niño intensity, and further research is needed to assess the extent of this modulation. The under-investigated relationship between TCs and ENSO could lead to an improved understanding of ENSO and help improve ENSO amplitude forecasts. How this relationship is incorporated into existing forecasting models still needs to be determined. Finally, these results suggest that the climatic effect of TC activity should be given more attention, including its effects on other climatic events.”

2. The corresponding explanations to reviewer’s possible doubt

2.1 Comparing with single-factor models (N3.4-model and ACE-model)

We mentioned “... that the modulated extent of WNP TCs to El Niño intensity may reach about 20%.”, it seems still a bit speculative, actually, we have proved it in our next work. A new physically based empirical model for ENSO prediction model has been built using the preceding (three months earlier) WNP ACE and N3.4.

Based on the possible physical mechanisms that link WNP TCs to ENSO, a new physically based empirical model for ENSO (the ACE+N3.4 model for short) is constructed. The prediction skill of the ACE+N3.4 model is assessed by comparison with single-factor models (the N3.4 and ACE models) and current dynamical and statistical models.

A holdout method is first employed. And the N3.4 predicted by models is

compared with observations There is a high correlation coefficient (~ 0.91) and sign consistency (92%) between the observed N3.4 and that predicted by the ACE+N3.4 model, far higher than is given by the single-factor N3.4 model (correlation coefficient of 0.75, sign consistency of 79%) and ACE model (correlation coefficient of 0.66, sign consistency of 65%) (Fig. A1a). The root mean square error (RMSE) between the observed N3.4 and that predicted by the ACE+N3.4 model is 0.39°C , which is smaller than that for the N3.4 model (0.58°C) and the ACE model (0.68°C). These results indicate that the ACE+N3.4 model has better prediction skill for N3.4 than single-factor models. Because the ACE model has the lowest prediction skill, we only consider the ACE+N3.4 model and the N3.4 model in further analyses of El Niño and La Niña events. The N3.4 predicted by the ACE+N3.4 model is closer to observations than that predicted by the N3.4 model during the development periods of El Niño and La Niña events (Figs. A1c and A1e). The peak of predictions by the N3.4 model occurs three months after observations, on the contrary, the predictions by the ACE+N3.4 model and observations simultaneously reach peak. **The maximum relative advantage of the ACE+N3.4 model for N3.4 relative to the N3.4 model is about 0.62°C in November during the El Niño developing year. This value of relative advantage is about 43% of the observed anomaly. Although the relative advantage of the ACE+N3.4 model relative to the N3.4-model for N3.4 in other months is smaller than in November, the proportion of value of relative advantage in observation might be higher in some months, such as in April, May, June, and July.** The relative predicted advantage of the ACE+N3.4 model is also evident during the La Niña developing year, with a maximum of nearly 0.5°C in December, this value of maximum relative advantage is about 46% of observation. This proportion might be proportionally higher in October and November. The predictions by the ACE+N3.4 and N3.4 models are similar during the decaying periods of both types of event, because there are few TCs in this period. **To evaluate further the predicted ability, a running holdout method is employed. The prediction skill of the ACE+N3.4 model for N3.4 is better than that with the holdout method (Fig. A1b, A1d and A1f). It does not seem to be very different.**

From these results, we can find the modulated extent of WNP TCs to El Niño intensity may exceeds 20% in some periods. Hence, in order to remain consistent with Fig. 1 in current manuscript, we said “the modulated extent of WNP TCs to El Niño intensity may reach about 20%.”

Fig. A1 Prediction skill of three statistical models for N3.4 in the hindcasting period 2000–2016. (a) Correlation coefficient, sign consistency (%) and RMSE ($^{\circ}\text{C}$) between N3.4 ($^{\circ}\text{C}$) observations and predictions based on three statistical models (N3.4 model, ACE model, and ACE+N3.4 model) using the holdout method. (b) Same as a, but for results using the running holdout method. (c) Composite time series of N3.4 for observations (black solid line) and predictions (dashed lines) during El Niño events with the holdout method. The blue (red) dashed line is for the N3.4 (ACE+N3.4) model. Bars indicate the amplitude of the models’ relative advantage (see **Methods**) between the N3.4 and ACE+N3.4 models. Blue (orange) bars represent the advantage of the N3.4 model relative to the ACE+N3.4 model (the ACE+N3.4 model relative to the N3.4 model). (d) Same as c, but for results using the running holdout method. (e) Same as c, but for La Niña. (f) Same as e, but for results using the running holdout method.

2.2 Comparing with current dynamics and statistical models

Model prediction skills for ENSO are compared for the ACE+N3.4 model, 18 dynamical models, and 8 current statistical models. For the whole year, the N3.4 predicted by the ACE+N3.4 model shows a high correlation (~ 0.92 ; Fig. A2a) and low RMSE ($\sim 0.35^{\circ}\text{C}$; Fig. A2b) with observations at a lead time of 2 months. The correlation coefficient (~ 0.91) and RMSE ($\sim 0.35^{\circ}\text{C}$) between the average skill level of dynamical models and observed N3.4 are nearly equivalent to those from the

ACE+N3.4 model, but the current statistical models clearly show a predicted disadvantage relative to the ACE+N3.4 model; the average correlation coefficient between prediction and observations is ~ 0.88 and the RMSE is $\sim 0.42^{\circ}\text{C}$. For the period July–December, the N3.4 predicted by the ACE+N3.4 model shows a higher correlation coefficient (~ 0.94 ; Fig. A2c) and lower RMSE ($\sim 0.32^{\circ}\text{C}$; Fig. A2d) than the average for both the dynamical models (correlation coefficient of ~ 0.91 , RMSE of $\sim 0.40^{\circ}\text{C}$) and pre-existing statistical models (correlation coefficient of ~ 0.90 , RMSE of $\sim 0.47^{\circ}\text{C}$). However, for the period January–June, the ACE+N3.4 model shows a predictability barrier, with its prediction skill for N3.4 being close to the average skill of the pre-existing statistical models (Figs. A2e and A2f). This might be related to the WNP TC genesis season, as there are few TCs in this period, which further verifies the importance of TCs in ENSO forecasts. El Niño and La Niña events are also examined (Figs. A3a and A3b). **Compared with the average prediction skill for current dynamical or statistical models, the ACE+N3.4 model has a higher prediction skill for ENSO intensity during the ENSO developing period. And both predicted N3.4 by the ACE+N3.4 model and observed N3.4 reaches peak at the same time, and the peak of average predictions from current dynamical or statistical models are both lagged.** This predicted advantage of the ACE+N3.4 model for ENSO is more evident when compared with the average prediction skill for current statistical models. The unexpected halt of N3.4 growth in 2014 and the development of an extreme El Niño in 2015 have attracted much research. Here, the predicted N3.4 values in 2014/2015 are shown. **Unlike the current dynamical and statistical models, the ACE+N3.4 model does not show the El Niño signal before October in 2014 (Fig. A3c) because of the anomalous absence of preceding TCs (Fig. A4a).** In 2014, the WNP ACE is far below the average ACE level during El Niño events, and is even close to the average WNP ACE during La Niña events. The N3.4 predicted by the ACE-model closely follows the variations in the observations after February of that year (Fig. A4b). **In 2015, the ACE+N3.4 model successfully predicts El Niño intensity close to the observations because the preceding WNP ACE in 2015 was well above the average ACE level of El Niño events.** During the 2015 El Niño developing period, the ACE+N3.4 model shows even some predicted advantage relative to the average skill of the dynamical models.

These comparisons of the ACE+N3.4, dynamical, and current statistical models amply confirm that the ACE+N3.4 model has a significant advantage in predicted ability for ENSO intensity, further verifying the importance of WNP

TCs to prediction of ENSO.

Fig. A2 Temporal correlation and RMSE between predictions and observations, as a function of lead time. (a) Correlation coefficient for all months combined. Each line represents one model. Eighteen dynamical models (thin dashed lines with circles), 8 statistical models (thin solid lines with asterisks), the N3.4 model (black thick line with dots) and the ACE+N3.4 model (red thick line with dots) are employed. The blue (green) thick line with dots denotes the average of the dynamical (statistical) models. **(b)** Same as **a**, but for RMSE. **(c)** Same as **a**, but for July–December. **(d)** Same as **c**, but for RMSE. **(e)** Same as **a**, but for January–June. **(f)** Same as **e**, but for RMSE.

Fig. A3 Time series of N3.4 (°C) for observations and predictions, together with observed ACE anomalies. (a) Composite time series of N3.4 for all El Niño events in the hindcasting period (2000–2016). The blue dashed line is the average prediction by dynamical models, and the red line the ACE+N3.4 model. The black solid line is the observations. Bars indicate the amplitude of the models' relative advantage (see **Methods**) between the average of the dynamical models and the ACE+N3.4 model. Blue (orange) bars represent the advantage of the dynamical models average relative to the ACE+N3.4 model (ACE+N3.4 model relative to dynamical models average). (b) Same as a, but for La Niña. (c) Time series of average N3.4 from dynamical models (blue dashed line), the ACE+N3.4 model (red dashed line), and observations (black solid line).

Fig. A4 Time series of the N3.4 (°C, 2014–2015) for observations and predictions, as well as observed ACE anomalies. (a) Time series of observed N3.4 and ACE anomalies. Black solid line is observed N3.4; red dotted line is observed ACE; green (blue) line is the composite of ACE anomalies during El Niño (La Niña). **(b)** Time series of average N3.4 from 2014 to 2015. Blue dashed line is the prediction from N3.4 model, green for ACE model; red solid line for ACE+N3.4 model, black for observations.

II. Response to Comments of Reviewer #2

Reviewer #2

The authors have done a commendable amount of work to address my earlier review. At the same time, my concerns about the causal chain being argued for have not been much alleviated. For example, Figure 3b clearly shows westerly winds on the equator before the TCs develop, at least as indicated by ACE, and I see no evidence that external influences such as the MJO have been ruled out. Beyond that, my earlier complaint that there are many rather speculative statements stands. For example, on lines 128-130, is stated that 'The main reason for this is a Hadley-like circulation (20° N–20°S, 135°–170°E) caused by the TCs'. How do we know that it was caused by the TCs?

It would be far better to try to publish this paper in a journal that specializes in ENSO-related research so that experts in the field can begin a discussion of the ideas presented here, before attempting to present it to a broader audience.

Response:

Thanks for the reviewer's helpful comments and suggestions. The manuscript has been revised carefully and more supporting materials have been added. More details and point-to-point responses to the reviewer's comments are listed as follows.

Q1. "Figure 3b clearly shows westerly winds on the equator before the TCs develop,

at least as indicated by ACE"

Response:

The occurrence of westerly winds disturbance on the south flank of TC is one of essential TCs genesis condition (Gray 1968, Emanuel 2003), hence, as the reviewers' saying, westerly winds near the equator occur before the TCs develop. In our study, we focus on the feedback of TCs to westerly wind. Actually, Figs. B1 and B2 are shown to explain this feedback. From Fig. B1 and B2, we can obtain three important messages

as follows:

1. The mean value of TC-related anomalous westerlies is larger than that without TC occurrence.
2. The center of anomalous westerlies varies with the movement of TCs.
3. The ratio between the mean anomalous westerlies with TCs and that without TCs holds regardless of whether El Niño happens, which means the role of single TC to westerlies doesn't change with different year.

All these results can prove that TCs have a positive feedback to westerly wind. We check situation in all years from 1970 to 2016. All situations accord with the aforementioned results.

Fig. B1. Latitude–time (along 135°–170°E, a) and longitude–time (along 0°–15°N, b) Hovmöller diagrams of zonal wind anomalies (shading, $m s^{-1}$) and WNP ACE ($m^2 s^{-2}$) in 2015. Black contours indicate the $1 m^2 s^{-2}$ isoline of WNP ACE, representing the TC life cycle.

Fig. B2. Regional meridionally-averaged (a) and zonally-averaged (b) values of the maximum zonal wind anomalies (m s^{-1}), and the ratios of mean values between the maximum zonal wind anomalies with TCs and that without TCs.

In addition, good prediction skill is most useful proof to our current study. In our next work, a new physically based empirical model for ENSO prediction model has been built on the basis of the preceding (three months earlier) WNP ACE and N3.4.

Based on the possible physical mechanisms that link WNP TCs to ENSO, a new physically based empirical model for ENSO (the ACE+N3.4 model for short) is constructed. The prediction skill of the ACE+N3.4 model is assessed by comparison with single-factor models (the N3.4 and ACE models) and current dynamical and statistical models:

1) Comparing with single-factor models

A holdout method is first employed. And the N3.4 predicted by models is compared with observations. There is a high correlation coefficient (~ 0.91) and sign consistency (92%) between the observed N3.4 and that predicted by the ACE+N3.4 model, far higher than is given by the single-factor N3.4 model (correlation coefficient of 0.75, sign consistency of 79%) and ACE model (correlation coefficient of 0.66, sign consistency of 65%) (Fig. B3a). The root mean square error (RMSE) between the observed N3.4 and that predicted by the ACE+N3.4 model is 0.39°C , which is smaller than that for the N3.4 model (0.58°C) and the ACE model (0.68°C). These results indicate that the ACE+N3.4 model has better prediction skill for N3.4 than single-factor models. Because the ACE model has the lowest prediction skill, we only consider the ACE+N3.4 model and the N3.4 model in further analyses of El Niño and La Niña events. The N3.4 predicted by the ACE+N3.4 model is closer to observations than that predicted by the N3.4 model during the development periods of El Niño and La Niña events (Figs. B3c and B3e). The peak of predictions by the N3.4 model occurs three months after observations, on the contrary, the predictions by the ACE+N3.4 model and

observations simultaneously reach peak. The maximum relative advantage of the ACE+N3.4 model for N3.4 relative to the N3.4 model is about 0.62°C in November during the El Niño developing year. This value of relative advantage is about 43% of the observed anomaly. Although the relative advantage of the ACE+N3.4 model relative to the N3.4-model for N3.4 in other months is smaller than in November, the proportion of value of relative advantage in observation might be higher in some months, such as in April, May, June, and July. The relative predicted advantage of the ACE+N3.4 model is also evident during the La Niña developing year, with a maximum of nearly 0.5°C in December, this value of maximum relative advantage is about 46% of observation. This proportion might be proportionally higher in October and November. The predictions by the ACE+N3.4 and N3.4 models are similar during the decaying periods of both types of event, because there are few TCs in this period. To evaluate further the predicted ability, a running holdout method is employed. The prediction skill of the ACE+N3.4 model for N3.4 is better than that with the holdout method (Fig. B3b, B3d and B3f). It does not seem to be very different.

Fig. B3 Prediction skill of three statistical models for N3.4 in the hindcasting period 2000–2016. (a) Correlation coefficient, sign consistency (%) and RMSE ($^{\circ}\text{C}$) between N3.4 ($^{\circ}\text{C}$) observations and predictions based on three statistical models (N3.4 model, ACE model, and

ACE+N3.4 model) using the holdout method. **(b)** Same as **a**, but for results using the running holdout method. **(c)** Composite time series of N3.4 for observations (black solid line) and predictions (dashed lines) during El Niño events with the holdout method. The blue (red) dashed line is for the N3.4 (ACE+N3.4) model. Bars indicate the amplitude of the models' relative advantage between the N3.4 and ACE+N3.4 models. Blue (orange) bars represent the advantage of the N3.4 model relative to the ACE+N3.4 model (the ACE+N3.4 model relative to the N3.4 model). **(d)** Same as **c**, but for results using the running holdout method. **(e)** Same as **c**, but for La Niña. **(f)** Same as **e**, but for results using the running holdout method.

2) Comparing with current dynamics and statistical models

Model prediction skills for ENSO are compared for the ACE+N3.4 model, 18 dynamical models, and 8 current statistical models. For the whole year, the N3.4 predicted by the ACE+N3.4 model shows a high correlation (~ 0.92 ; Fig. B4a) and low RMSE ($\sim 0.35^\circ\text{C}$; Fig. B4b) with observations at a lead time of 2 months. The correlation coefficient (~ 0.91) and RMSE ($\sim 0.35^\circ\text{C}$) between the average skill level of dynamical models and observed N3.4 are nearly equivalent to those from the ACE+N3.4 model, but the current statistical models clearly show a predicted disadvantage relative to the ACE+N3.4 model; the average correlation coefficient between prediction and observations is ~ 0.88 and the RMSE is $\sim 0.42^\circ\text{C}$. For the period July–December, the N3.4 predicted by the ACE+N3.4 model shows a higher correlation coefficient (~ 0.94 ; Fig. B4c) and lower RMSE ($\sim 0.32^\circ\text{C}$; Fig. B4d) than the average for both the dynamical models (correlation coefficient of ~ 0.91 , RMSE of $\sim 0.40^\circ\text{C}$) and pre-existing statistical models (correlation coefficient of ~ 0.90 , RMSE of $\sim 0.47^\circ\text{C}$). However, for the period January–June, the ACE+N3.4 model shows a predictability barrier, with its prediction skill for N3.4 being close to the average skill of the pre-existing statistical models (Figs. B4e and B4f). This might be related to the WNP TC genesis season, as there are few TCs in this period, which further verifies the importance of TCs in ENSO forecasts. El Niño and La Niña events are also examined (Figs. B5a and B5b). **Compared with the average prediction skill for current dynamical or statistical models, the ACE+N3.4 model has a higher prediction skill for ENSO intensity during the ENSO developing period. And both predicted N3.4 by the ACE+N3.4 model and observed N3.4 reaches peak at the same time, and the peak of average predictions from current dynamical or statistical models are both**

lagged. This predicted advantage of the ACE+N3.4 model for ENSO is more evident when compared with the average prediction skill for current statistical models. The unexpected halt of N3.4 growth in 2014 and the development of an extreme El Niño in 2015 have attracted much research. Here, the predicted N3.4 values in 2014/2015 are shown. **Unlike the current dynamical and statistical models, the ACE+N3.4 model does not show the El Niño signal before October in 2014 (Fig. B5c) because of the anomalous absence of preceding TCs (Fig. B6a).** In 2014, the WNP ACE is far below the average ACE level during El Niño events, and is even close to the average WNP ACE during La Niña events. The N3.4 predicted by the ACE-model closely follows the variations in the observations after February of that year (Fig. B6b). **In 2015, the ACE+N3.4 model successfully predicts El Niño intensity close to the observations because the preceding WNP ACE in 2015 was well above the average ACE level of El Niño events.** During the 2015 El Niño developing period, the ACE+N3.4 model shows even some predicted advantage relative to the average skill of the dynamical models.

These comparisons of the ACE+N3.4, dynamical, and current statistical models amply confirm that the ACE+N3.4 model has a significant advantage in predicted ability for ENSO intensity, further verifying the importance of WNP TCs to prediction of ENSO.

References :

- Emanuel, K., 2003: Tropical cyclones. *Annu Rev Earth Pl Sc*, **31**: 75-104.
- Gray, W. M., 1968: Global view of origin of tropical disturbances and storms. *Mon Weather Rev*, **96**: 669-&.

Fig. B4 Temporal correlation and RMSE between predictions and observations, as a function of lead time. (a) Correlation coefficient for all months combined. Each line represents one model. Eighteen dynamical models (thin dashed lines with circles), 8 statistical models (thin solid lines with asterisks), the N3.4 model (black thick line with dots) and the ACE+N3.4 model (red thick line with dots) are employed. The blue (green) thick line with dots denotes the average of the dynamical (statistical) models. **(b)** Same as **a**, but for RMSE. **(c)** Same as **a**, but for July–December. **(d)** Same as **c**, but for RMSE. **(e)** Same as **a**, but for January–June. **(f)** Same as **e**, but for RMSE.

Fig. B5 Time series of N3.4 (°C) for observations and predictions, together with observed ACE anomalies. (a) Composite time series of N3.4 for all El Niño events in the hindcasting period (2000–2016). The blue dashed line is the average prediction by dynamical models, and the red line the ACE+N3.4 model. The black solid line is the observations. Bars indicate the amplitude of the models' relative advantage (see **Methods**) between the average of the dynamical models and the ACE+N3.4 model. Blue (orange) bars represent the advantage of the dynamical models average relative to the ACE+N3.4 model (ACE+N3.4 model relative to dynamical models average). (b) Same as a, but for La Niña. (c) Time series of average N3.4 from dynamical models (blue dashed line), the ACE+N3.4 model (red dashed line), and observations (black solid line).

Fig. B6 Time series of the N3.4 (°C, 2014–2015) for observations and predictions, as well as observed ACE anomalies. (a) Time series of observed N3.4 and ACE anomalies. Black solid line is observed N3.4; red dotted line is observed ACE; green (blue) line is the composite of ACE anomalies during El Niño (La Niña). **(b)** Time series of average N3.4 from 2014 to 2015. Blue dashed line is the prediction from N3.4 model, green for ACE model; red solid line for ACE+N3.4 model, black for observations.

Q2. “and I see no evidence that external influences such as the MJO have been ruled out”

Response:

Thanks your good questions. Fig. B7 shows the lead-lag correlation between the monthly N3.4 and ACE. Firstly, when the SST data from the ERA Interim dataset is employed (in above analysis, monthly SST is from ERSST V5 dataset of NOAA), the result that the WNP ACE is strongly correlated with Niño 3.4 (N3.4) three months later is robust (Figs. B7a and B7b). Secondly, as can be seen from Figs B7c and B7d, the lead–lag correlation between WNP ACE and N3.4 is largely unaffected by removal of the MJO signal, and the magnitude of the correlation coefficient only changes slightly. That is, this MJO signal might hardly affect the modulation of TCs to ENSO intensity on interannual timescale. As shown in Fig. B8a, compared with original time series, when the MJO is removed, the intensity of N3.4 is barely changed, it’s also true to ACE (Fig. B8b). What’s more, the intensity of N3.4 related to the preceding ACE (after removing MJO signal) is also hardly changed (Fig. B8c). This shows that the MJO signal might only slightly affect the modulation of TCs to ENSO intensity on interannual timescale. In our previous manuscripts, because of the inaccurate

expression, we may make a misunderstanding that MJO has no influence to this modulation. Hence, we change all expression in the related part (lines:197-198, 203-204, 208-209, 215-216). Also, this two figures have been added in the manuscript (Extended Data Figs. 10 and 11, lines:191-199).

Fig. B7 Lead-lag correlations between WNP ACE index and N3.4 and their autocorrelations in the period 1970–2016. **a**, Lead-lag correlation and autocorrelations between original series. The red, green and blue dashed lines indicate significant at the 99% confidence level related to lead-lag correlations between WNP ACE index and N3.4, autocorrelations of N3.4 and WNP ACE index via Student’s *t*-test using the effective number of degrees of freedom (see **Methods**), respectively. **b**, Lead-lag correlations between the processed series (see **Methods**). $N3.4_{N3.4_0}$ ($ACE_{N3.4_0}$) indicates the N3.4 (WNP ACE index) that is not associated with the preceding (three months earlier) N3.4. $ACE_{N3.4^*}$ indicates the WNP ACE index that is not associated with the simultaneous N3.4. The red, green and blue dashed lines indicate significant at the 99% confidence level related to lead-lag correlations between $N3.4$ and $ACE_{N3.4_0}$, $N3.4$ and $ACE_{N3.4^*}$, $N3.4_{N3.4_0}$ and $ACE_{N3.4_0}$ via Student’s *t*-test using the effective number of degrees of freedom (see **Methods**), respectively., **(c–d)**, same as **(a–b)**, but for WNP ACE index and N3.4 (both removing MJO).

Fig. B8 Time series of the preceding (3-months earlier) WNP ACE ($\text{m}^2 \text{s}^{-2}$) and N3.4 ($^{\circ}\text{C}$), and regressions onto the N3.4 from 1970 to 2016. **a**, Preceding N3.4. N3.4_0 denotes the original series, and N3.4_0^* indicates the series not associated with the MJO index. **b**, Preceding ACE. ACE_0 denotes the original series, and ACE_0^* indicates the series not associated with the preceding MJO. **c**, Regression on the N3.4. N3.4_ACE_0 denotes the regression of the preceding ACE index on the N3.4, and N3.4_ACE_0^* indicates the regression of the preceding ACE index not associated with the MJO on the N3.4.

Q3. *“on lines 128-130, is stated that 'The main reason for this is a Hadley-like circulation (20°N – 20°S , 135° – 170°E) caused by the TCs'. How do we know that it was caused by the TCs?”*

Response:

Thanks for your good question. As can be seen from Fig. B9, during the El Niño developing year, there is anomalous updrafts in 10° – 20°N , where is the key domain of ACE. Meantime, anomalous downdrafts occur in the southern hemisphere. After removing ACE signal, anomalous vertical circulation disappears.

Fig. B9. Composite of the vertical p -velocity (shading) and the wind (vectors) in the vertical–meridional plane (m s^{-1}) in J-A-S in the period of 1970–2016. Shading and black vectors indicate significant above the 95% confidence level using Student’s t -test. (a), Climatic wind field. (b), Wind field during El Niño developing years. (c), Wind field after removing the J-A-S ACE during El Niño developing years

What’s more, we explore the composite 200 hPa meridional geopotential height gradient anomalies in J-A-S associated with the J-A-S WNP ACE during El Niño developing years (Fig. B10). In the northern hemisphere, there is a positive potential height gradient anomalies on the southern flanks of WNP TCs at 200 hPa. On the contrary, a negative potential height gradient anomalies occurs in the southern hemisphere. According to the tropical semi-geostrophic adjustment, it’s can explain the symmetry of the wind field in the upper troposphere with respect to the equator. We have added this part to manuscript.

Fig. B10. Composite 200 hPa meridional geopotential height gradient anomalies (10^{-5} gpm m^{-1}) in J-A-S associated with the J-A-S WNP ACE during El Niño developing years. The stippled regions show significant above the 95% confidence level using Student's t -test. The black rectangle denotes the selected ACE region (10° – 20° N, 135° – 170° E)

Q4. *“It would be far better to try to publish this paper in a journal that specializes in ENSO-related research so that experts in the field can begin a discussion of the ideas presented here, before attempting to present it to a broader audience.”*

Response:

Thanks for your warm reminder. As the reviewer's saying, *Nature Communication* has wider audience than other special journal, it includes not only ENSO-related experts, but also TC-related experts. All researches need a wide discussion before estimating its value, we'd like to accept advices and doubts of experts from any field. Also, in our next work, we have verified that TCs can improve the current ENSO predictions indeed. These results also remind us that the climatic effect of TC activities has been ignored for a long time, it should be given more attention, even on other climatic events. *Nature Communication* has not only a broader audience also more experts in all fields, which is what we expect.

More materials

Thank for reviewer's all questions, they gave us many great reminders. Except for some aforementioned evidences, we also added the detailed discussion for the limitations in the manuscript (lines: 226-243):

Although the primary focus here is on the feedback of WNP TC occurrence on westerlies and the thermocline, other related topics warrant further investigation. For example, the effect of the MJO and other factors on TC genesis and the possible physical mechanisms associated with the influence of WNP TCs on Hadley-like circulations and tropical westerlies still need to be verified in a fully coupled model. Previous studies^{18, 26, 30} have provided fundamental insights into simulating climatic TCs in a fully coupled model, but were restricted to TC climatic distributions (or to

accumulated TCs). Currently, fully coupled models that perform well in simulating interannual variabilities of both TC activity and El Niño are rare, and remain an important challenge for researchers. Furthermore, the observed explained percentage of N3.4 intensity during the El Niño developing years in this work (Fig. 1) shows that the modulated strength of WNP TCs to El Niño intensity may reach ~20%. The absence of TCs can weaken El Niño intensity, and further research is needed to assess the extent of this modulation. The under-investigated relationship between TCs and ENSO could lead to an improved understanding of ENSO and help improve ENSO amplitude forecasts. How this relationship is incorporated into existing forecasting models still needs to be determined. Finally, these results suggest that the climatic effect of TC activity should be given more attention, including its effects on other climatic events.

I. Response to Comments of Reviewer #1

Reviewer #1

The manuscript is improved. The ideas are interesting and fit with recent literature (though they go into much more depth here). Conclusions are still a bit speculative, and I suggest adding in some text on caveats/limitations based on the reviewers' points. I also suggest a careful edit for English. I recommend acceptance after these minor recommendations.

Response:

1. Caveats/limitations

Thanks for the reviewer's warm reminder, as the reviewer's saying, more text on caveats/limitations should be added in the closing discussion paragraph indeed. We have divided the original closing paragraph to two paragraphs: the key results in the first part and a detailed discussion including limitations in the second part.

The first part (lines: 301-325 in the manuscript):

Firstly, the feedback of WNP TCs on El Niño on interannual timescale is examined using observational data over several decades in this work. It is found that WNP TCs can significantly intensify El Niño 3-months later, particularly for J-A-S TCs. This modulation has few dependencies on the MJO. WNP TCs in J-A-S can affect both the atmospheric and oceanic processes that drive the O-N-D El Niño (Fig. 10), which is different from that during the early Pliocene epoch¹⁸. WNP TCs can weaken the Walker circulation via direct effects of equatorial asymmetrically anomalous westerlies at lower tropospheric levels and indirect effects of Hadley-like circulation. TCs can shallow the thermocline in the tropical Western Pacific. Warm water in the tropical Western Pacific is carried eastward, in association with the enhanced eastward-

propagating equatorial Kelvin waves, further deepening the thermocline in the tropical eastern Pacific, thereby reducing the gradient of the zonal thermocline in the equatorial Pacific. These two processes lead to intensified El Niño and weakened La Niña. The greater the WNP ACE, the stronger (weaker) the El Niño (La Niña).

Secondly, a new physics-based empirical model (the ACE+N3.4 model) has been constructed based on the preceding ACE and N3.4. Compared with current dynamical and statistical models, the ACE+N3.4 model, although very simple, gives a significantly better ENSO intensity forecast, especially for June–December during the developing period of ENSO. Meantime, it's evident that the predicted N3.4 peak by current dynamical and statistical models show the lagged signal compared to observation (the “target period slippage” of ENSO¹⁴), but for the ACE+N3.4 model, the predictions and observations concurrently reach peak. The TC genesis season in the WNP directly affects the forecasting ability of the ACE+N3.4 model. The predictions of the ACE+N3.4 model also provide powerful evidence that TCs are essential to ENSO development and can improve significantly the prediction skill for ENSO intensity.

The second part (lines: 326-339 in the manuscript):

Although the primary focus here is on the feedback of WNP TC occurrence on ENSO, other related topics warrant further investigation. For example, the effect of the MJO and other factors on TC genesis and the possible physical mechanisms associated with the influence of WNP TCs on Hadley-like circulations and tropical westerlies still need to be verified in a fully coupled model. Previous studies^{18, 26, 30} have provided fundamental insights into simulating climatic TCs in a fully coupled model, but were restricted to TC climatic distributions (or to accumulated TCs). Currently, fully coupled models that perform well in simulating interannual variabilities of both TC activity and El Niño are rare, and remain an important challenge for researchers. In general, this under-investigated relationship between TCs and ENSO could lead to an improved understanding of ENSO and help improve ENSO amplitude forecasts. How this relationship is incorporated into existing forecasting models still needs to be determined. Finally, these results suggest that the climatic effect of TC activity should be given more attention, including its effects on other climatic events.

2. A new physics-based empirical model for ENSO prediction

Good prediction skill is most useful proof to our current study. In this work, a new physically based empirical model for ENSO prediction model has been built on the basis of the preceding (three months earlier) WNP ACE and N3.4.

Based on the possible physical mechanisms that link WNP TCs to ENSO, a new physically based empirical model for ENSO (the ACE+N3.4 model for short) is constructed. The prediction skill of the ACE+N3.4 model is assessed by comparison with single factor models (the N3.4 and ACE models) and current dynamical and statistical models (details are shown in lines: 216-299 in the manuscript):

1) Comparing with single factor models

A holdout method is first employed. And the N3.4 predicted by models is compared with observations. There is a high correlation coefficient (~ 0.91) and sign consistency (92%) between the observed N3.4 and that predicted by the ACE+N3.4 model, far higher than is given by the single-factor N3.4 model (correlation coefficient of 0.75, sign consistency of 79%) and ACE model (correlation coefficient of 0.66, sign consistency of 65%) (Fig. A1a). The root mean square error (RMSE) between the observed N3.4 and that predicted by the ACE+N3.4 model is 0.39°C , which is smaller than that for the N3.4 model (0.58°C) and the ACE model (0.68°C). These results indicate that the ACE+N3.4 model has better prediction skill for N3.4 than single factor models. Because the ACE model has the lowest prediction skill, we only consider the ACE+N3.4 model and the N3.4 model in further analyses of El Niño and La Niña events. The N3.4 predicted by the ACE+N3.4 model is closer to observations than that predicted by the N3.4 model during the development periods of El Niño and La Niña events (Figs. A1c and A1e). The peak of predictions by the N3.4 model occurs three months after observations, on the contrary, the predictions by the ACE+N3.4 model and observations simultaneously reach peak. **The maximum relative advantage of the ACE+N3.4 model for N3.4 relative to the N3.4 model is about 0.62°C in November during the El Niño developing year. This value of relative advantage is about 43% of the observed anomaly. Although the relative advantage of the ACE+N3.4 model**

relative to the N3.4-model for N3.4 in other months is smaller than in November, the proportion of value of relative advantage in observation might be higher in some months, such as in April, May, June, and July. The relative predicted advantage of the ACE+N3.4 model is also evident during the La Niña developing year, with a maximum of nearly 0.5°C in December, this value of maximum relative advantage is about 46% of observation. This proportion might be proportionally higher in October and November. The predictions by the ACE+N3.4 and N3.4 models are similar during the decaying periods of both types of event, because there are few TCs in this period. **To evaluate further the predicted ability, a running holdout method is employed. The prediction skill of the ACE+N3.4 model for N3.4 is better than that with the holdout method (Fig. A1b). It does not seem to be very different.** The ACE+N3.4 model also shows better predicted ability for ENSO than the N3.4 model, especially during the development periods of El Niño and La Niña events (Figs. A1d and A1f). **This predicted ability does not change with the selection of the training period (Fig. A2);** the mean correlation coefficient between predicted and observed N3.4 is 0.912 and the mean RMSE is 0.376°C. This verifies the stability of the ACE+N3.4 model.

Fig. A1 Prediction skill of three statistical models for N3.4 in the hindcasting period 2000–2016. (a) Correlation coefficient, sign consistency (%) and RMSE ($^{\circ}\text{C}$) between N3.4 ($^{\circ}\text{C}$) observations and predictions based on three statistical models (N3.4 model, ACE model, and ACE+N3.4 model) using the holdout method. (b) Same as a, but for results using the running holdout method. (c) Composite time series of N3.4 for observations (black solid line) and predictions (dashed lines) during El Niño events with the holdout method. The blue (red) dashed line is for the N3.4 (ACE+N3.4) model. Bars indicate the amplitude of the models' relative advantage (see **Methods**) between the N3.4 and ACE+N3.4 models. Blue (orange) bars represent the advantage of the N3.4 model relative to the ACE+N3.4 model (the ACE+N3.4 model relative to the N3.4 model). (d) Same as c, but for results using the running holdout method. (e) Same as c, but for La Niña. (f) Same as e, but for results using the running holdout method.

Fig. A2 Correlation coefficient and RMSE (°C) between observed and predicted N3.4 from ACE+N3.4 model using the LPO-CV (see Methods).

2) Comparing with current dynamics and statistical models

Model prediction skills for ENSO are compared for the ACE+N3.4 model, 18 dynamical models, and 8 current statistical models. For the whole year, the N3.4 predicted by the ACE+N3.4 model shows a high correlation (~ 0.92 ; Fig. A3a) and low RMSE ($\sim 0.35^\circ\text{C}$; Fig. A3b) with observations at a lead time of 2 months. The correlation coefficient (~ 0.91) and RMSE ($\sim 0.35^\circ\text{C}$) between the average skill level of dynamical models and observed N3.4 are nearly equivalent to those from the ACE+N3.4 model, but the current statistical models clearly show a predicted disadvantage relative to the ACE+N3.4 model; the average correlation coefficient between prediction and observations is ~ 0.88 and the RMSE is $\sim 0.42^\circ\text{C}$. For the period July–December, the N3.4 predicted by the ACE+N3.4 model shows a higher correlation coefficient (~ 0.94 ; Fig. A3c) and lower RMSE ($\sim 0.32^\circ\text{C}$; Fig. A3d) than the average for both the dynamical models (correlation coefficient of ~ 0.91 , RMSE of $\sim 0.40^\circ\text{C}$) and pre-existing statistical models (correlation coefficient of ~ 0.90 , RMSE of $\sim 0.47^\circ\text{C}$). However, for the period January–June, the ACE+N3.4 model shows a predictability barrier, with its prediction skill for N3.4 being close to the average skill of the pre-existing statistical models (Figs. A3e and A3f). This might be related to the WNP TC genesis season, as there are few TCs in this period, which further verifies the importance of TCs in ENSO forecasts. El Niño and La Niña events are also examined (Figs. A4a and A4b). **Compared with the average prediction skill for current dynamical or statistical models, the ACE+N3.4 model has a higher prediction skill for ENSO intensity during the ENSO developing period. And both predicted N3.4 by the ACE+N3.4 model and observed N3.4 reaches peak at the same time, and the peak**

of average predictions from current dynamical or statistical models are both lagged. This predicted advantage of the ACE+N3.4 model for ENSO is more evident when compared with the average prediction skill for current statistical models. The unexpected halt of N3.4 growth in 2014 and the development of an extreme El Niño in 2015 have attracted much research. Here, the predicted N3.4 values in 2014/2015 are shown. **Unlike the average prediction skill of current dynamical and statistical models, the ACE+N3.4 model does not show the El Niño signal before October in 2014 (Fig. A4c) because of the anomalous absence of preceding TCs (Fig.A5a).** In 2014, the WNP ACE is far below the average ACE level during El Niño events, and is even close to the average WNP ACE during La Niña events. The N3.4 predicted by the ACE-model closely follows the variations in the observations after February of that year (Fig. A5b). **In 2015, the ACE+N3.4 model successfully predicts El Niño intensity close to the observations because the preceding WNP ACE in 2015 was well above the average ACE level of El Niño events.** During the 2015 El Niño developing period, the ACE+N3.4 model shows even some predicted advantage relative to the average skill of the dynamical models.

These comparisons of the ACE+N3.4, dynamical, and current statistical models amply confirm that the ACE+N3.4 model has a significant advantage in predicted ability for ENSO intensity, further verifying the importance of WNP TCs to prediction of ENSO.

Fig. A3 Temporal correlation and RMSE between predictions and observations, as a function of lead time. (a) Correlation coefficient for all months combined. Each line represents one model. Eighteen dynamical models (thin dashed lines with circles), 8 statistical models (thin solid lines with asterisks), the N3.4 model (black thick line with dots) and the ACE+N3.4 model (red thick line with dots) are employed. The blue (green) thick line with dots denotes the average of the dynamical (statistical) models. (b) Same as a, but for RMSE. (c) Same as a, but for July–December. (d) Same as c, but for RMSE. (e) Same as a, but for January–June. (f) Same as e, but for RMSE.

Fig. A4 Time series of N3.4 (°C) for observations and predictions, together with observed ACE anomalies. (a) Composite time series of N3.4 for all El Niño events in the hindcasting period (2000–2016). The blue dashed line is the average prediction by dynamical models, and the red line the ACE+N3.4 model. The black solid line is the observations. Bars indicate the amplitude of the models' relative advantage (see **Methods**) between the average of the dynamical models and the ACE+N3.4 model. Blue (orange) bars represent the advantage of the dynamical models average relative to the ACE+N3.4 model (ACE+N3.4 model relative to dynamical models average). (b) Same as a, but for La Niña. (c) Time series of average N3.4 from dynamical models (blue dashed line), the ACE+N3.4 model (red dashed line), and observations (black solid line).

Fig. A5 Time series of the N3.4 (°C, 2014–2015) for observations and predictions, as well as observed ACE anomalies. (a) Time series of observed N3.4 and ACE anomalies. Black solid line is observed N3.4; red dotted line is observed ACE; green (blue) line is the composite of ACE anomalies during El Niño (La Niña). **(b)** Time series of average N3.4 from 2014 to 2015. Blue dashed line is the prediction from N3.4 model, green for ACE model; red solid line for ACE+N3.4 model, black for observations.

II. Response to Comments of Reviewer #2

Reviewer #2

The authors have done a commendable amount of work to address my earlier review. At the same time, my concerns about the causal chain being argued for have not been much alleviated. For example, Figure 3b clearly shows westerly winds on the equator before the TCs develop, at least as indicated by ACE, and I see no evidence that external influences such as the MJO have been ruled out. Beyond that, my earlier complaint that there are many rather speculative statements stands. For example, on lines 128-130, is stated that 'The main reason for this is a Hadley-like circulation (20° N–20°S, 135°–170°E) caused by the TCs'. How do we know that it was caused by the TCs?

It would be far better to try to publish this paper in a journal that specializes in ENSO-related research so that experts in the field can begin a discussion of the ideas presented here, before attempting to present it to a broader audience.

Response:

Thanks for the reviewer's helpful comments and suggestions. The manuscript has been revised carefully and more supporting materials have been added. More details and point-to-point responses to the reviewer's comments are listed as follows.

Q1. "Figure 3b clearly shows westerly winds on the equator before the TCs develop,

at least as indicated by ACE"

Response:

The occurrence of westerly winds disturbance on the south flank of TC is one of

essential TCs genesis condition (Gray 1968, Emanuel 2003), hence, as the reviewers' saying, westerly winds near the equator occur before the TCs develop. In our study, we focus on the feedback of TCs to westerly wind. Actually, Figs. B1 and B2 are shown to explain this feedback. From Fig. B1 and B2, we can obtain three important messages as follows:

1. The mean value of TC-related anomalous westerlies is larger than that without TC occurrence.
2. The center of anomalous westerlies varies with the movement of TCs.
3. The ratio between the mean anomalous westerlies with TCs and that without TCs holds regardless of whether El Niño happens, which means the role of single TC to westerlies doesn't change with different year.

All these results can prove that TCs have a positive feedback to westerly wind. We check situation in all years from 1970 to 2016. All situations accord with the aforementioned results.

Fig. B1. Latitude–time (along 135°–170°E, a) and longitude–time (along 0°–15°N, b) Hovmöller diagrams of zonal wind anomalies (shading, $m s^{-1}$) and WNP ACE ($m^2 s^{-2}$) in 2015. Black contours indicate the 1 $m^2 s^{-2}$ isoline of WNP ACE, representing the TC life cycle.

Fig. B2. Regional meridionally-averaged (a) and zonally-averaged (b) values of the maximum zonal wind anomalies (m s^{-1}), and the ratios of mean values between the maximum zonal wind anomalies with TCs and that without TCs.

In addition, good prediction skill is most useful proof to our current study. In this work, a new physically based empirical model for ENSO prediction model has been built on the basis of the preceding (three months earlier) WNP ACE and N3.4.

Based on the possible physical mechanisms that link WNP TCs to ENSO, a new physically based empirical model for ENSO (the ACE+N3.4 model for short) is constructed. The prediction skill of the ACE+N3.4 model is assessed by comparison with single-factor models (the N3.4 and ACE models) and current dynamical and statistical models (details are shown in lines: 216-299 in the manuscript):

1) Comparing with single factor models

A holdout method is first employed. And the N3.4 predicted by models is compared with observations. There is a high correlation coefficient (~ 0.91) and sign consistency (92%) between the observed N3.4 and that predicted by the ACE+N3.4 model, far higher than is given by the single-factor N3.4 model (correlation coefficient of 0.75, sign consistency of 79%) and ACE model (correlation coefficient of 0.66, sign consistency of 65%) (Fig. B3a). The root mean square error (RMSE) between the observed N3.4 and that predicted by the ACE+N3.4 model is 0.39°C , which is smaller than that for the N3.4 model (0.58°C) and the ACE model (0.68°C). These results indicate that the ACE+N3.4 model has better prediction skill for N3.4 than single-factor models. Because the ACE model has the lowest prediction skill, we only consider the ACE+N3.4 model and the N3.4 model in further analyses of El Niño and La Niña events.

The N3.4 predicted by the ACE+N3.4 model is closer to observations than that predicted by the N3.4 model during the development periods of El Niño and La Niña events (Figs. B3c and B3e). The peak of predictions by the N3.4 model occurs three months after observations, on the contrary, the predictions by the ACE+N3.4 model and observations simultaneously reach peak. **The maximum relative advantage of the ACE+N3.4 model for N3.4 relative to the N3.4 model is about 0.62°C in November during the El Niño developing year. This value of relative advantage is about 43% of the observed anomaly. Although the relative advantage of the ACE+N3.4 model relative to the N3.4-model for N3.4 in other months is smaller than in November, the proportion of value of relative advantage in observation might be higher in some months, such as in April, May, June, and July.** The relative predicted advantage of the ACE+N3.4 model is also evident during the La Niña developing year, with a maximum of nearly 0.5°C in December, this value of maximum relative advantage is about 46% of observation. This proportion might be proportionally higher in October and November. The predictions by the ACE+N3.4 and N3.4 models are similar during the decaying periods of both types of event, because there are few TCs in this period. **To evaluate further the predicted ability, a running holdout method is employed. The prediction skill of the ACE+N3.4 model for N3.4 is better than that with the holdout method (Fig. B3b). It does not seem to be very different.** The ACE+N3.4 model also shows better predicted ability for ENSO than the N3.4 model, especially during the development periods of El Niño and La Niña events (Figs. B3d and B3f). **This predicted ability does not change with the selection of the training period (Fig. B4);** the mean correlation coefficient between predicted and observed N3.4 is 0.912 and the mean RMSE is 0.376°C. This verifies the stability of the ACE+N3.4 model.

Fig. B3 Prediction skill of three statistical models for N3.4 in the hindcasting period 2000–2016. (a) Correlation coefficient, sign consistency (%) and RMSE (°C) between N3.4 (°C) observations and predictions based on three statistical models (N3.4 model, ACE model, and ACE+N3.4 model) using the holdout method. (b) Same as a, but for results using the running holdout method. (c) Composite time series of N3.4 for observations (black solid line) and predictions (dashed lines) during El Niño events with the holdout method. The blue (red) dashed line is for the N3.4 (ACE+N3.4) model. Bars indicate the amplitude of the models' relative advantage between the N3.4 and ACE+N3.4 models. Blue (orange) bars represent the advantage of the N3.4 model relative to the ACE+N3.4 model (the ACE+N3.4 model relative to the N3.4 model). (d) Same as c, but for results using the running holdout method. (e) Same as c, but for La Niña. (f) Same as e, but for results using the running holdout method.

Fig. B4 Correlation coefficient and RMSE (°C) between observed and predicted N3.4 from ACE+N3.4 model using the LPO-CV (see Methods).

2) Comparing with current dynamics and statistical models

Model prediction skills for ENSO are compared for the ACE+N3.4 model, 18 dynamical models, and 8 current statistical models. For the whole year, the N3.4 predicted by the ACE+N3.4 model shows a high correlation (~ 0.92 ; Fig. B5a) and low RMSE ($\sim 0.35^\circ\text{C}$; Fig. B5b) with observations at a lead time of 2 months. The correlation coefficient (~ 0.91) and RMSE ($\sim 0.35^\circ\text{C}$) between the average skill level of dynamical models and observed N3.4 are nearly equivalent to those from the ACE+N3.4 model, but the current statistical models clearly show a predicted disadvantage relative to the ACE+N3.4 model; the average correlation coefficient between prediction and observations is ~ 0.88 and the RMSE is $\sim 0.42^\circ\text{C}$. For the period July–December, the N3.4 predicted by the ACE+N3.4 model shows a higher correlation coefficient (~ 0.94 ; Fig. B5c) and lower RMSE ($\sim 0.32^\circ\text{C}$; Fig. B5d) than the average for both the dynamical models (correlation coefficient of ~ 0.91 , RMSE of $\sim 0.40^\circ\text{C}$) and pre-existing statistical models (correlation coefficient of ~ 0.90 , RMSE of $\sim 0.47^\circ\text{C}$). However, for the period January–June, the ACE+N3.4 model shows a predictability barrier, with its prediction skill for N3.4 being close to the average skill of the pre-existing statistical models (Figs. B5e and B5f). This might be related to the WNP TC genesis season, as there are few TCs in this period, which further verifies the importance of TCs in ENSO forecasts. El Niño and La Niña events are also examined (Figs. B6a and B6b). **Compared with the average prediction skill for current dynamical or statistical models, the ACE+N3.4 model has a higher prediction skill for ENSO intensity during the ENSO developing period. And both predicted N3.4 by the ACE+N3.4 model and observed N3.4 reaches peak at the same time, and the peak of average predictions from current dynamical or statistical models are both lagged.** This predicted advantage of the ACE+N3.4 model for ENSO is more evident when compared with the average prediction skill for current statistical models. The unexpected halt of N3.4 growth in 2014 and the development of an extreme El Niño in 2015 have attracted much research. Here, the predicted N3.4 values in 2014/2015 are shown. **Unlike the average prediction skill for current dynamical and statistical**

models, the ACE+N3.4 model does not show the El Niño signal before October in 2014 (Fig. B6c) because of the anomalous absence of preceding TCs (Fig.B7a). In 2014, the WNP ACE is far below the average ACE level during El Niño events, and is even close to the average WNP ACE during La Niña events. The N3.4 predicted by the ACE-model closely follows the variations in the observations after February of that year (Fig. B7b). **In 2015, the ACE+N3.4 model successfully predicts El Niño intensity close to the observations because the preceding WNP ACE in 2015 was well above the average ACE level of El Niño events.** During the 2015 El Niño developing period, the ACE+N3.4 model shows even some predicted advantage relative to the average skill of the dynamical models.

These comparisons of the ACE+N3.4, dynamical, and current statistical models amply confirm that the ACE+N3.4 model has a significant advantage in predicted ability for ENSO intensity, further verifying the importance of WNP TCs to prediction of ENSO.

References :

- Emanuel, K., 2003: Tropical cyclones. *Annu Rev Earth Pl Sc*, **31**: 75-104.
- Gray, W. M., 1968: Global view of origin of tropical disturbances and storms. *Mon Weather Rev*, **96**: 669-&.

Fig. B5 Temporal correlation and RMSE between predictions and observations, as a function of lead time. (a) Correlation coefficient for all months combined. Each line represents one model. Eighteen dynamical models (thin dashed lines with circles), 8 statistical models (thin solid lines with asterisks), the N3.4 model (black thick line with dots) and the ACE+N3.4 model (red thick line with dots) are employed. The blue (green) thick line with dots denotes the average of the dynamical (statistical) models. **(b)** Same as **a**, but for RMSE. **(c)** Same as **a**, but for July–December. **(d)** Same as **c**, but for RMSE. **(e)** Same as **a**, but for January–June. **(f)** Same as **e**, but for RMSE.

Fig. B6 Time series of N3.4 (°C) for observations and predictions, together with observed ACE anomalies. (a) Composite time series of N3.4 for all El Niño events in the hindcasting period (2000–2016). The blue dashed line is the average prediction by dynamical models, and the red line the ACE+N3.4 model. The black solid line is the observations. Bars indicate the amplitude of the models' relative advantage (see **Methods**) between the average of the dynamical models and the ACE+N3.4 model. Blue (orange) bars represent the advantage of the dynamical models average relative to the ACE+N3.4 model (ACE+N3.4 model relative to dynamical models average). (b) Same as a, but for La Niña. (c) Time series of average N3.4 from dynamical models (blue dashed line), the ACE+N3.4 model (red dashed line), and observations (black solid line).

Fig. B7 Time series of the N3.4 (°C, 2014–2015) for observations and predictions, as well as observed ACE anomalies. (a) Time series of observed N3.4 and ACE anomalies. Black solid line is observed N3.4; red dotted line is observed ACE; green (blue) line is the composite of ACE anomalies during El Niño (La Niña). **(b)** Time series of average N3.4 from 2014 to 2015. Blue dashed line is the prediction from N3.4 model, green for ACE model; red solid line for ACE+N3.4 model, black for observations.

Q2. “and I see no evidence that external influences such as the MJO have been ruled out”

Response:

Thanks your good questions. Fig. B7 shows the lead-lag correlation between the monthly N3.4 and ACE. Firstly, when the SST data from the ERA Interim dataset is employed (in above analysis, monthly SST is from ERSST V5 dataset of NOAA), the result that the WNP ACE is strongly correlated with Niño 3.4 (N3.4) three months later is robust (Figs. B8a and B8b). Secondly, as can be seen from Figs B8c and B8d, the lead–lag correlation between WNP ACE and N3.4 is largely unaffected by removal of the MJO signal, and the magnitude of the correlation coefficient only changes slightly. That is, this MJO signal might hardly affect the modulation of TCs to ENSO intensity on interannual timescale. As shown in Fig. B9a, compared with original time series, when the MJO is removed, the intensity of N3.4 is barely changed, it’s also true to ACE (Fig. B9b). What’s more, the intensity of N3.4 related to the preceding ACE (after

removing MJO signal) is also hardly changed (Fig. B9c). This shows that the MJO signal might only slightly affect the modulation of TCs to ENSO intensity on interannual timescale. In our previous manuscripts, because of the inaccurate expression, we may make a misunderstanding that MJO has no influence to this modulation. Hence, we change all expression in the related part (lines:201-203, 207-208, 212-213, 303-304). Also, this two figures have been added in the manuscript (Extended Data Figs. 10 and 11, lines:195-201).

Fig. B8 Lead-lag correlations between WNP ACE index and N3.4 and their autocorrelations in the period 1970–2016. **a**, Lead-lag correlation and autocorrelations between original series. The red, green and blue dashed lines indicate significant at the 99% confidence level related to lead-lag correlations between WNP ACE index and N3.4, autocorrelations of N3.4 and WNP ACE index via Student’s *t*-test using the effective number of degrees of freedom (see **Methods**), respectively. **b**, Lead-lag correlations between the processed series (see **Methods**). $N3.4_{N3.4_0}^*$ ($ACE_{N3.4_0}^*$) indicates the N3.4 (WNP ACE index) that is not associated with the preceding (three months earlier) N3.4. $ACE_{N3.4}^*$ indicates the WNP ACE index that is not associated with the simultaneous N3.4. The red, green and blue dashed lines indicate significant at the 99% confidence level related to lead-lag correlations between $N3.4$ and $ACE_{N3.4_0}^*$, $N3.4$ and $ACE_{N3.4}^*$, $N3.4_{N3.4_0}^*$ and $ACE_{N3.4_0}^*$ via Student’s *t*-test using the effective number of degrees of freedom (see **Methods**), respectively., **(c–d)**, same as **(a–b)**, but for WNP ACE index and N3.4 (both removing MJO).

Fig. B9 Time series of the preceding (3-months earlier) WNP ACE ($\text{m}^2 \text{s}^{-2}$) and N3.4 ($^{\circ}\text{C}$), and regressions onto the N3.4 from 1970 to 2016. **a**, Preceding N3.4. N3.4_0 denotes the original series, and N3.4_0^* indicates the series not associated with the MJO index. **b**, Preceding ACE. ACE_0 denotes the original series, and ACE_0^* indicates the series not associated with the preceding MJO. **c**, Regression on the N3.4. N3.4_ACE_0 denotes the regression of the preceding ACE index on the N3.4, and N3.4_ACE_0^* indicates the regression of the preceding ACE index not associated with the MJO on the N3.4.

Q3. *“on lines 128-130, is stated that 'The main reason for this is a Hadley-like circulation (20°N – 20°S , 135° – 170°E) caused by the TCs'. How do we know that it was caused by the TCs?”*

Response:

Thanks for your good question. As can be seen from Fig. B10, during the El Niño developing year, there is anomalous updrafts in 10° – 20°N , where is the key domain of

ACE. Meantime, anomalous downdrafts occur in the southern hemisphere. After removing ACE signal, anomalous vertical circulation disappears.

Fig. B10. Composite of the vertical p -velocity (shading) and the wind (vectors) in the vertical-meridional plane (m s^{-1}) in J-A-S in the period of 1970–2016. Shading and black vectors indicate significant above the 95% confidence level using Student’s t -test. (a), Climatic wind field. (b), Wind field during El Niño developing years. (c), Wind field after removing the J-A-S ACE during El Niño developing years

What’s more, we explore the composite 200 hPa meridional geopotential height gradient anomalies in J-A-S associated with the J-A-S WNP ACE during El Niño developing years (Fig. B11). In the northern hemisphere, there is a positive potential height gradient anomalies on the southern flanks of WNP TCs at 200 hPa. On the contrary, a negative potential height gradient anomalies occurs in the southern hemisphere. According to the tropical semi-geostrophic adjustment, it’s can explain the symmetry of the wind field in the upper troposphere with respect to the equator. We have added this part to manuscript (Extended Data Fig. 6b in the manuscript).

Fig. B11. Composite 200 hPa meridional geopotential height gradient anomalies (10^{-5} gpm m^{-1}) in J-A-S associated with the J-A-S WNP ACE during El Niño developing years. The stippled regions show significant above the 95% confidence level using Student's t -test. The black rectangle denotes the selected ACE region (10° – 20° N, 135° – 170° E)

Q4. *“It would be far better to try to publish this paper in a journal that specializes in ENSO-related research so that experts in the field can begin a discussion of the ideas presented here, before attempting to present it to a broader audience.”*

Response:

Thanks for your warm reminder. As the reviewer's saying, *Nature Communication* has wider audience than other special journal, it includes not only ENSO-related experts, but also TC-related experts. All researches need a wide discussion before estimating its value, we'd like to accept advices and doubts of experts from any field. Also, in this work, we have verified that TCs can improve the current ENSO predictions indeed. These results also remind us that the climatic effect of TC activities has been ignored for a long time, it should be given more attention, even on other climatic events. *Nature Communication* has not only a broader audience also more experts in all fields, which is what we expect.

More materials

Thank for reviewer's all questions, they gave us many great reminders. Except for some aforementioned evidences, we also added the detailed discussion for the limitations in the manuscript (lines: 326-339):

Although the primary focus here is on the feedback of WNP TC occurrence on ENSO, other related topics warrant further investigation. For example, the effect of the MJO and other factors on TC genesis and the possible physical mechanisms associated with the influence of WNP TCs on Hadley-like circulations and tropical westerlies still need to be verified in a fully coupled model. Previous studies^{18, 26, 30} have provided fundamental insights into simulating climatic TCs in a fully coupled model, but were restricted to TC climatic distributions (or to accumulated TCs). Currently, fully coupled models that perform well in simulating interannual variabilities of both TC activity and El Niño are rare, and remain an important challenge for researchers. In general, this under-investigated relationship between TCs and ENSO could lead to an improved understanding of ENSO and help improve ENSO amplitude forecasts. How this relationship is incorporated into existing forecasting models still needs to be determined. Finally, these results suggest that the climatic effect of TC activity should be given more attention, including its effects on other climatic events.

Reviewers' comments:

Reviewer #1 (Remarks to the Author):

The authors addressed my minor concerns. The science is interesting and noteworthy, and I recommend publication after improvements to the presentation.

I suggest revising the text for flow and grammar, and add more specific text on implications for climate and prediction... avoid over general statements such as the last sentence of the text relating the importance of the "climatic effect of TCs" on "other climatic events" . What does this even mean? The results have wide-ranging applications in large-scale dynamics and weather/climate interactions and predictability. I suggest the authors add more text highlighting these broader impacts of the work for a multi-disciplinary audience.

Reviewer #2 (Remarks to the Author):

The authors have responded well to my comments on their revised paper, though I still have some reservations about statements on causality made on the basis of lagged correlations. It is time to let readers decide if it is plausible that tropical cyclones affect the progression of ENSO.

I. Response to Comments of Reviewer #1

Reviewer #1

The authors addressed my minor concerns. The science is interesting and noteworthy, and I recommend publication after improvements to the presentation.

I suggest revising the text for flow and grammar, and add more specific text on implications for climate and prediction... avoid over general statements such as the last sentence of the text relating the importance of the "climatic effect of TCs" on "other climatic events". What does this even mean? The results have wide-ranging applications in large-scale dynamics and weather/climate interactions and predictability. I suggest the authors add more text highlighting these broader impacts of the work for a multi-disciplinary audience.

Response:

We sincerely thank reviewer for the time and effort in reviewing our manuscript. Firstly, we have revised the English language of the whole manuscript by asking a professional company, *Stallard Scientific Editing*. Secondly, according to the reviewer's reminder about the wide-ranging applications and implications of this work in the corresponding aspects, we have added the more detailed description in the discussion (lines:358-375 in the manuscript):

“In general, although TC activity is usually considered a synoptic event, the cumulative effect of TCs can provide a significant cross-scale feedback to the large-scale atmospheric and oceanic circulations that then affects the intensity of ENSO. TC activity thus help to improve ENSO forecasts. How this under-investigated relationship between TCs and ENSO is incorporated into existing forecasting models still needs to be determined. In addition, these results highlight the need for further study of the cross-scale effect of short-timescale events on long-timescale events. Perhaps the cumulative effect of TCs in other basins (e.g., hurricanes over the North Atlantic) affects intraseasonal or longer-timescale climatic events in other regions by modulating the large-scale dynamic processes. The cumulative effect may be a critical link between synoptic and climatic events. If this cumulative effect of short-term events can be reproduced well in the existing forecasting models, the prediction skill of models would be improved. Furthermore, we can also apply this cumulative effect to cross-scale studies in other disciplines. For example, in oceanography, it may deepen our understanding of the inverse energy cascade from mesoscale eddies to large-scale circulation. All these aspects belong to the category of cross-scale dynamics, which deserves further in-depth study.”

II. Response to Comments of Reviewer #2

Reviewer #2

The authors have responded well to my comments on their revised paper, though I still have some reservations about statements on causality made on the basis of lagged correlations. It is time to let readers decide if it is plausible that tropical cyclones affect the progression of ENSO.

Response:

We sincerely thank reviewer for the time and effort in reviewing our manuscript. In order to show our results more objectively to the readers, we have given a clearer claim about limitation of causality in the discussion (lines: 350-354 in the manuscript):

“Although the primary focus here is on the feedback of WNP TC occurrence on ENSO, other related topics warrant further investigation. For example, the effect of the MJO and other factors on TC genesis and the causality associated with the influence of WNP TCs on Hadley-like circulations and tropical westerlies still need to be further verified in a fully coupled model.”

In addition, we asked a professional company, *Stallard Scientific Editing*, to improve English.